# Amortized Inference of Causal Models via Conditional Fixed-Point Iterations

## Abstract

Structural Causal Models (SCMs) offer a principled framework to reason about interventions and support out-of-distribution generalization, which are key goals in scientific discovery. However, the task of learning SCMs from observed data poses formidable challenges, and often requires training a separate model for each dataset. In this work, we propose amortized inference of SCMs by training a *single* model on multiple datasets sampled from different SCMs. We first use a transformer-based architecture for amortized learning of dataset embeddings, and then extend the Fixed-Point Approach (FiP) [Scetbon et al., 2024] to infer SCMs conditionally on their dataset embeddings. As a byproduct, our method can generate observational and interventional data from novel SCMs at inference time, without updating parameters. Empirical results show that our amortized procedure performs on par with baselines trained specifically for each dataset on both in and out-of-distribution problems, and also outperforms them in scare data regimes.

## 1 Introduction

Learning structural causal models (SCMs) from observations is a core problem in many scientific domains [Sachs et al., 2005, Foster et al., 2011, Xie et al., 2012], as SCMs provide a principled way to model the data generation process. They enable simulation of controlled interventions, offering the potential to accelerate scientific discovery by predicting the outcomes of unseen experiments without requiring costly/time-consuming lab trials [Ke et al., 2023, Zhang et al., 2024]. However, solving this inverse problem of learning SCMs from observed data is challenging as both the causal graph and the causal mechanisms are unknown a priori. Recovering causal graphs is an NP-hard combinatorial optimization problem as the space of causal graphs is super-exponential [Chickering et al., 2004]. This subsequently complicates the estimation of causal mechanisms via maximum likelihood estimation per node [Blöbaum et al., 2022]. To address these challenges, recent approaches have focused on learning causal mechanisms with partial causal structure, using techniques such as autoregressive flows [Khemakhem et al., 2021, Geffner et al., 2022, Javaloy et al., 2023], or modeling SCMs as fixed-point iterations via transformers [Scetbon et al., 2024].

Despite these advances, a major limitation remains: each new dataset requires training a specific model, that prevents the transfer of causal knowledge across datasets. Amortized inference offers a solution by learning a *single* model that can generalize across instances of the same optimization problem by exploiting their shared structure [Andrychowicz et al., 2016, Gordon et al., 2019]. This results in models that can quickly adapt to new instances at test time [Finn et al., 2017]. Amortized inference has been shown success in several challenging tasks, like bayesian posterior estimation [Garnelo et al., 2018, Müller et al., 2021], sampling from unnormalized densities [Akhound-Sadegh et al., 2024, Sendera et al., 2024], as well as causal structure learning [Lorch et al., 2022, Ke et al., 2022], which is more aligned with our paper.

Submitted to 39th Conference on Neural Information Processing Systems (NeurIPS 2025). Do not distribute.

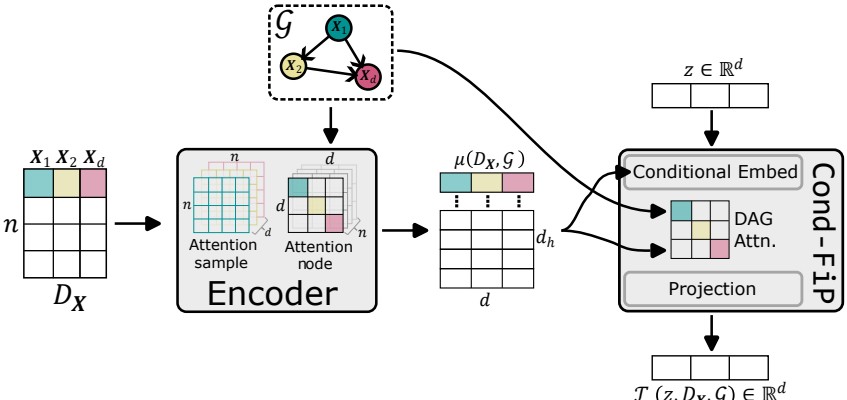

Figure 1: Sketch of the approach proposed in this work. Given a dataset of observations $D_{\boldsymbol{X}}$ and a causal graph $\mathcal{G}$ obtained from an unknown SCM $\mathcal{S}(\mathbb{P}_{\boldsymbol{N}}, \mathcal{G}, \boldsymbol{F})$, the encoder produces a dataset embedding $\mu(D_{\boldsymbol{X}}, \mathcal{G})$, which serves as a condition to instantiate Cond-FiP. Then for any point $\boldsymbol{z} \in \mathbb{R}^d$, $\mathcal{T}(\boldsymbol{z}, D_{\boldsymbol{X}}, \mathcal{G})$ aims at replicating the functional mechanism $\boldsymbol{F}(\boldsymbol{z})$ of the generative SCM.

In this work, we tackle the novel problem of amortized inference of causal mechanisms for additive noise SCMs. We propose a two-step approach where we first learn dataset embeddings via in-context learning [Garg et al., 2022] to represent the task-specific information. These embeddings are then used to condition the fixed-point (FiP) approach [Scetbon et al., 2024] for modeling causal mechanisms. This conditional modification, termed *Cond-FiP*, enables the model to adapt the causal mechanism for each specific instance (Figure 1). Our key contributions are highlighted below.

- We propose Cond-FiP, a novel extension of FiP approach that enables amortized inference by training a single model across different instances from the functional class of SCMs.

- For novel SCMs at inference, Cond-FiP can recover the causal mechanisms from the input observations without updating any parameters, thereby allowing us to generate observational and interventional data on the fly.

- We show empirically that Cond-FiP achieves similar performances as the state-of-the-art (SOTA) approaches trained from scratch for each dataset on both in and out-of-distribution (OOD) problems. Further, Cond-FiP obtains better results than baselines in scare data regimes, due to its amortized inference procedure.

## 2 Amortized Causal Learning

### 2.1 Brief Overview of Amortized Inference

Amortized inference aims to learn a shared inference mechanism across multiple tasks that enables fast adaptation to new tasks at test time. Consider a task $T$ that defines a distribution over inputs ($\boldsymbol{Z}$) and targets ($\boldsymbol{Y}$), i.e, $\boldsymbol{Z}, \boldsymbol{Y} \sim \mathbb{P}_T$. Given a collection of tasks $\left(T^{(k)}\right)_{k=1}^K$ and some objective function $L$, the goal is to learn a model $\mathcal{T}_\theta$ shared across tasks as follows:

$$\arg\min_\theta \sum_k \mathbb{E}_{\boldsymbol{Z}, \boldsymbol{Y} \sim \mathbb{P}_{T^{(k)}}} L(\boldsymbol{Y}, \mathcal{T}_\theta(\boldsymbol{Z}, \boldsymbol{I}^{(k)})) \tag{1}$$

where $\boldsymbol{I}^{(k)}$ denotes additional context for task $T^{(k)}$, such as dataset with samples $[\boldsymbol{Z}_1, \cdots, \boldsymbol{Z}_n]$. Instead of retraining from scratch, the model should leverage the context $\boldsymbol{I}'$ to adapt to the task $T'$.

A classic approach for this is meta-learning [Andrychowicz et al., 2016, Finn et al., 2017], that utilizes context $\boldsymbol{I}'$ by task-specific finetuning. These methods typically learn a shared initialization that is refined for a specific task via few gradient steps in an inner optimization loop.

In contrast, in-context learning (ICL) [Müller et al., 2021, Xie et al., 2021, Garg et al., 2022] avoids this inner loop by using transformer-based architectures. By attending to the context $\boldsymbol{I}'$ during the

forward pass, ICL methods adapt to a specific task without any parameter updates. This ability arises from the observation that transformers can implicitly approximate learning algorithms such as gradient descent within their activation dynamics [Akyürek et al., 2022, Von Oswald et al., 2023].

## 2.2 Problem Setup

We start by formally defining structural causal models (SCMs). An SCM defines the causal generative process of a set of $d$ endogenous (causal) random variables $\boldsymbol{V} = \{X_1, \cdots, X_d\}$, where each causal variable $X_i$ is defined as a function of a subset of other causal variables ($\boldsymbol{V} \setminus \{X_i\}$) and an exogenous noise variable $N_i$:

$$X_i = F_i(PA(X_i), N_i) \text{ s.t. } PA(X_i) \subset \boldsymbol{V} , X_i \notin PA(X_i) \tag{2}$$

Hence, an SCM $\mathcal{S}(\mathbb{P}_{\boldsymbol{N}}, \boldsymbol{\mathcal{G}}, \boldsymbol{F})$ describes the data-generation process of $\boldsymbol{X} := [X_1, \cdots, X_d] \sim \mathbb{P}_{\boldsymbol{X}}$ from the noise variables $\boldsymbol{N} := [N_1, \cdots, N_d] \sim \mathbb{P}_{\boldsymbol{N}}$ via the function $\boldsymbol{F} := [F_1, \cdots, F_d]$, and a graph $\boldsymbol{\mathcal{G}} \in \{0,1\}^{d \times d}$ indicating the parents of each $X_i$, that is $[\boldsymbol{\mathcal{G}}]_{i,j} := 1$ if $X_j \in PA(X_i)$. We make the following assumptions about SCMs.

- $\boldsymbol{\mathcal{G}}$ is a directed and acyclic graph (DAG), and noise variables are mutually independent (Markovian SCM).
- SCMs are restricted to be *additive noise models* (ANM), i.e., $X_i = F_i(PA(X_i)) + N_i$.

While the first assumption is pretty standard, we make the ANM assumption for training the proposed dataset encoder in Section 3.1.

Consider a distribution over SCMs $\mathcal{S}(\mathbb{P}_{\boldsymbol{N}}, \boldsymbol{\mathcal{G}}, \boldsymbol{F}) \sim \mathbb{P}_{\mathcal{S}}$. Then the goal with amortized inference of causal mechanisms is to learn a single model $\mathcal{T}_\theta$ that can approximate the true causal mechanism $\boldsymbol{F}(\boldsymbol{z})$ for any input $\boldsymbol{z} \in \mathbb{R}^d$. With task specific context as $\boldsymbol{I} = (D_{\mathbf{X}}, \boldsymbol{\mathcal{G}})$ in equation 1, we have

$$\arg\min_\theta \mathbb{E}_{\mathcal{S} \sim \mathbb{P}_{\mathcal{S}}} \mathbb{E}_{\boldsymbol{z} \sim \mathbb{P}_{\boldsymbol{X}}} L(\boldsymbol{F}(\boldsymbol{z}), \mathcal{T}_\theta(\boldsymbol{z}, D_{\mathbf{X}}, \boldsymbol{\mathcal{G}})) \tag{3}$$

Note that we consider access to causal graph $\boldsymbol{\mathcal{G}}$ as part of the input context, which is available when training on synthetic SCMs. Even if we don't have access to $\boldsymbol{\mathcal{G}}$, we can use prior works on amortized causal learning [Lorch et al., 2022, Ke et al., 2022] to infer the causal graphs from observations $D_{\boldsymbol{X}}$. This justifies our setup where the causal graphs are provided as part of the context to the model.

## 3 Methodology: Conditional FiP

We present our methodology for learning the model $\mathcal{T}(., D_{\boldsymbol{X}}, \boldsymbol{\mathcal{G}})$ that consists of two components: (1) a dataset encoder that generates dataset embeddings $\mu(D_{\boldsymbol{X}}, \boldsymbol{\mathcal{G}})$ from the input context, and (2) a conditional variant of FiP [Scetbon et al., 2024], termed Cond-FiP that allows it to leverage the task-specific context for amortized inference via the learned dataset embeddings $\mu(D_{\boldsymbol{X}}, \boldsymbol{\mathcal{G}})$. We first present our dataset encoder, then Cond-FiP, and conclude with data generation via Cond-Fip.

## 3.1 Dataset Encoder

The objective of this section is to develop a method capable of producing efficient latent representations of datasets. To achieve this, we propose to train an encoder that predicts the noise samples from their associated observations given the causal structures via in-context learning.

**Training Setting.** We consider empirical representations of $K$ SCMs $\left(\mathcal{S}(\mathbb{P}_{\boldsymbol{N}}^{(k)}, \boldsymbol{\mathcal{G}}^{(k)}, \boldsymbol{F}^{(k)})\right)_{k=1}^K$, each sampled independently from a distribution over SCMs $\mathcal{S}(\mathbb{P}_{\boldsymbol{N}}^{(k)}, \boldsymbol{\mathcal{G}}^{(k)}, \boldsymbol{F}^{(k)}) \sim \mathbb{P}_{\mathcal{S}}$. Each empirical representation, denoted $(D_{\boldsymbol{X}}^{(k)}, \boldsymbol{\mathcal{G}}^{(k)})_{k=1}^K$, contains $n$ observations $D_{\boldsymbol{X}}^{(k)} := [\boldsymbol{X}_1^{(k)}, \ldots, \boldsymbol{X}_n^{(k)}]^T \in \mathbb{R}^{n \times d}$, and the causal graph $\boldsymbol{\mathcal{G}}^{(k)} \in \{0,1\}^{d \times d}$. For training, we also need the associated noise samples $D_{\boldsymbol{N}}^{(k)} := [\boldsymbol{N}_1^{(k)}, \ldots, \boldsymbol{N}_n^{(k)}]^T \in \mathbb{R}^{n \times d}$, which play the role of the target variable in our prediction task. For simplicity, we drop the index $k$ in our notation and assume access to the full distribution $\mathbb{P}_{\mathcal{S}}$. The objective is to recover the true noise $D_{\boldsymbol{N}}$ from a dataset of observations $D_{\boldsymbol{X}}$ and the causal graph $\boldsymbol{\mathcal{G}}$, which provide us with dataset embeddings as detailed below.

**Encoder Architecture.** Following [Lorch et al., 2021, Scetbon et al., 2024], we encode datasets using a transformer-based architecture that alternates attention over both sample and node dimension. Given a dataset $D_{\boldsymbol{X}}$, we first apply a linear embedding $L(D_{\boldsymbol{X}}) \in \mathbb{R}^{n \times d \times d_h}$, where $d_h$ is the hidden dimension. The encoder $E$ then applies transformer blocks, each comprising self-attention followed by an MLP [Vaswani et al., 2017], where the attention mechanism is applied either across the samples $n$ or the nodes $d$ alternately. Recall the standard self-attention is defined as

$$\mathrm{A}_{\boldsymbol{M}}(\boldsymbol{Q}, \boldsymbol{K}) = \frac{\exp((\boldsymbol{Q}\boldsymbol{K}^T - \boldsymbol{M})/\sqrt{d_h})}{\exp((\boldsymbol{Q}\boldsymbol{K}^T - \boldsymbol{M})/\sqrt{d_h})\, \mathbf{1}_d}$$

where $\boldsymbol{Q}, \boldsymbol{K} \in \mathbb{R}^{d \times d_h}$ denote the keys and queries for a single attention head, and $\boldsymbol{M} \in \{0, +\infty\}^{d \times d}$ is a (potential) mask. When attending over samples, the encoder uses standard self-attention without masking ($\boldsymbol{M} = \{0\}^{n \times n}$). But for node-wise attention, we incorporate causal structure by masking invalid dependencies using mask $\boldsymbol{M} = +\infty \times (1 - \boldsymbol{\mathcal{G}})$ in standard self-attention, with the convention that $0 \times (+\infty) = 0$. Finally, the embeddings $E(L(D_{\boldsymbol{X}}), \boldsymbol{\mathcal{G}}) \in \mathbb{R}^{n \times d \times d_h}$ are passed to a prediction network $H : \mathbb{R}^{n \times d \times d_h} \to \mathbb{R}^{n \times d}$, implemented as 2-hidden layers MLP to project back to the original data space.

**Training Procedure.** We minimize the mean squared error (MSE) of predicting the target $D_{\boldsymbol{N}}$ from the input $(D_{\boldsymbol{X}}, \boldsymbol{\mathcal{G}})$ over the distribution of SCMs $\mathbb{P}_{\mathcal{S}}$ available during training:

$$\mathbb{E}_{\mathcal{S} \sim \mathbb{P}_{\mathcal{S}}} \|D_{\boldsymbol{N}} - H \circ E(L(D_{\boldsymbol{X}}), \boldsymbol{\mathcal{G}})\|_2^2 .$$

Further, as we restrict ourselves to the case of ANMs, we can equivalently reformulate our training objective in order to predict the causal mechanism rather than the noise samples, as $\boldsymbol{F}(D_{\boldsymbol{X}}) := D_{\boldsymbol{X}} - D_{\boldsymbol{N}}$. Therefore, we instead propose to train our encoder as follows:

$$\mathbb{E}_{\mathcal{S} \sim \mathbb{P}_{\mathcal{S}}} \|\boldsymbol{F}(D_{\boldsymbol{X}}) - H \circ E(L(D_{\boldsymbol{X}}), \boldsymbol{\mathcal{G}})\|_2^2 .$$

Note that ANM assumption provides a simplified true mapping from data to noise as $x \to x - F(x)$, which is difficult to obtain in general SCMs. Please check Appendix A.2 for more details on justification for ANMs and why recovering noise is equivalent to learning the inverse SCM.

**Inference.** Given a new dataset $D_{\boldsymbol{X}}$ and its causal graph $\boldsymbol{\mathcal{G}}$, encoder provides us with the dataset embedding $\mu(D_{\boldsymbol{X}}, \boldsymbol{\mathcal{G}}) := E(L(D_{\boldsymbol{X}}), \boldsymbol{\mathcal{G}}) \in \mathbb{R}^{n \times d \times d_h}$.

## 3.2 Cond-FiP: Conditional Fixed-Point Decoder

We now present the modification of FiP that uses the learned dataset embeddings $\mu(D_{\boldsymbol{X}}, \boldsymbol{\mathcal{G}})$ for amortized inference of causal mechanisms.

**Training Setting.** Analogous to the encoder training setup, we assume that we have access to a distribution of SCMs $\mathcal{S}(\mathbb{P}_{\boldsymbol{N}}, \boldsymbol{\mathcal{G}}, \boldsymbol{F}) \sim \mathbb{P}_{\mathcal{S}}$ at training time, from which we can extract empirical representations $(D_{\boldsymbol{X}}, \boldsymbol{\mathcal{G}})$. Our goal is to train $\mathcal{T}$ such that given the context $(D_{\boldsymbol{X}}, \boldsymbol{\mathcal{G}})$ from an SCM $\mathcal{S}(\mathbb{P}_{\boldsymbol{N}}, \boldsymbol{\mathcal{G}}, \boldsymbol{F}) \sim \mathbb{P}_{\mathcal{S}}$, the induced conditional function $\boldsymbol{z} \in \mathbb{R}^d \to \mathcal{T}(\boldsymbol{z}, D_{\boldsymbol{X}}, \boldsymbol{\mathcal{G}}) \in \mathbb{R}^d$ approximates the true causal mechanisms $\boldsymbol{F} : \boldsymbol{z} \in \mathbb{R}^d \to \boldsymbol{F}(\boldsymbol{z}) \in \mathbb{R}^d$ (E.q. 3).

**Decoder Architecture.** The design of our decoder is based on the FiP architecture for fixed-point SCM learning, with two major differences: (1) we use the dataset embeddings $\mu(D_{\boldsymbol{X}}, \boldsymbol{\mathcal{G}})$ as a high dimensional codebook to embed the nodes, and (2) we leverage adaptive layer norm operators [Peebles and Xie, 2023] in the transformer blocks of FiP to enable conditional attention mechanisms.

**Conditional Embedding.** The key change of our decoder compared to the original FiP is in the embedding of the input. FiP proposes to embed a data point $\boldsymbol{z} := [z_1, \ldots, z_d] \in \mathbb{R}^d$ into a high dimensional space using a learnable codebook $\boldsymbol{C} := [C_1, \ldots, C_d]^T \in \mathbb{R}^{d \times d_h}$ and positional embedding $\boldsymbol{P} := [P_1, \ldots, P_d]^T \in \mathbb{R}^{d \times d_h}$, from which they define:

$$\boldsymbol{z}_{\mathrm{emb}} := [z_1 * C_1, \ldots, z_d * C_d]^T + \boldsymbol{P} \in \mathbb{R}^{d \times d_h}$$

This ensures that the embedded samples preserve the original causal structure. However, this embedding layer is only adapted if the samples considered are all drawn from the same observational distribution, as the representation of the nodes given by the codebook $\boldsymbol{C}$, is fixed. In order to

generalize their embedding strategy to the case where multiple SCMs are considered, we consider
conditional codebooks and positional embeddings adapted for each dataset. Given a dataset $D_{\boldsymbol{X}}$ and
a causal graph $\boldsymbol{\mathcal{G}}$, we propose to define the conditional codebook and positional embedding as

$$\boldsymbol{C}(D_{\boldsymbol{X}}, \boldsymbol{\mathcal{G}}) := \mu(D_{\boldsymbol{X}}, \boldsymbol{\mathcal{G}})W_{\boldsymbol{C}}$$
$$\boldsymbol{P}(D_{\boldsymbol{X}}, \boldsymbol{\mathcal{G}}) := \mu(D_{\boldsymbol{X}}, \boldsymbol{\mathcal{G}})W_{\boldsymbol{P}}$$

where $\mu(D_{\boldsymbol{X}}, \boldsymbol{\mathcal{G}}) := \mathrm{MaxPool}(E(L(D_{\boldsymbol{X}}), \boldsymbol{\mathcal{G}})) \in \mathbb{R}^{d \times d_h}$ is obtained by max-pooling w.r.t the sample
dimension the dataset embedding $E(L(D_{\boldsymbol{X}}), \boldsymbol{\mathcal{G}}) \in \mathbb{R}^{n \times d \times d_h}$ produced by our trained encoder, and
$W_{\boldsymbol{C}}, W_{\boldsymbol{P}} \in \mathbb{R}^{d_h \times d_h}$ are learnable parameters. Then we propose to embed any point $\boldsymbol{z} \in \mathbb{R}^d$
conditionally on the context $(D_{\boldsymbol{X}}, \boldsymbol{\mathcal{G}})$ as follows:

$$\boldsymbol{z}_{\mathrm{emb}} := [z_1 * C_1(D_{\boldsymbol{X}}, \boldsymbol{\mathcal{G}}), \ldots, z_d * C_d(D_{\boldsymbol{X}}, \boldsymbol{\mathcal{G}})]^T + \boldsymbol{P}(D_{\boldsymbol{X}}, \boldsymbol{\mathcal{G}}) \in \mathbb{R}^{d \times d_h}$$

**Adaptive Transfomer Block.** Once an input $\boldsymbol{z} \in \mathbb{R}^d$ has been embedded as $\boldsymbol{z}_{\mathrm{emb}} \in \mathbb{R}^{d \times d_h}$, FiP
models SCMs by simulating the reconstruction of the data from noise. Starting from $\boldsymbol{n}_0 \in \mathbb{R}^{d \times d_h}$ a
learnable parameter, they propose to update the current noise $L \geq 1$ times by computing:

$$\boldsymbol{n}_{\ell+1} = h(\mathrm{DA}_{\boldsymbol{M}}(\boldsymbol{n}_\ell, \boldsymbol{z}_{\mathrm{emb}})\boldsymbol{z}_{\mathrm{emb}} + \boldsymbol{n}_\ell)$$

where $h$ refers to the MLP block, and for clarity, we omit both the layer's dependence on its
parameters and the inclusion of layer normalization in the notation. Note that here FiP considers
the DAG-Attention mechanism (details in Appendix A.1) in order to correctly model the root
nodes of the SCM. To obtain a conditional formulation, we first replace the starting noise $\boldsymbol{n}_0$ with
$\boldsymbol{n}_0 := \mu(D_{\boldsymbol{X}}, \boldsymbol{\mathcal{G}})W_{\boldsymbol{n}_0} \in \mathbb{R}^{d \times d_h}$, where $W_{\boldsymbol{n}_0} \in \mathbb{R}^{d_h \times d_h}$ is a learnable parameter. Then we add
adaptive layer normalization operators [Peebles and Xie, 2023] to both attention and MLP blocks,
where each scale or shift is obtained by applying a 1 hidden-layer MLP to the embedding $\mu(D_{\boldsymbol{X}}, \boldsymbol{\mathcal{G}})$.

**Projection.** To project back the latent representation of $\boldsymbol{z}$ obtained from previous stages, $\boldsymbol{n}_L \in \mathbb{R}^{d \times d_h}$,
we use a linear operation to get $\widehat{\boldsymbol{z}} = \boldsymbol{n}_L W_{\mathrm{out}} \in \mathbb{R}^d$, where $W_{\mathrm{out}} \in \mathbb{R}^{d_h}$ is learnable.

**Training Procedure.** The result of forward pass can be summarized as $\widehat{\boldsymbol{z}} = \mathcal{T}(\boldsymbol{z}, D_{\boldsymbol{X}}, \boldsymbol{\mathcal{G}})$, where we
omit the dependence of $\widehat{\boldsymbol{z}}$ on context $(D_{\boldsymbol{X}}, \boldsymbol{\mathcal{G}})$ for simplicity. We train the model $\mathcal{T}$ by minimizing
the reconstruction error of the true causal mechanisms estimated by our model over the distribution
of SCMs $\mathbb{P}_{\mathcal{S}}$, as shown below.

$$\mathbb{E}_{\mathcal{S} \sim \mathbb{P}_{\mathcal{S}}} \mathbb{E}_{\boldsymbol{z} \sim \mathbb{P}_{\boldsymbol{X}}} \|\mathcal{T}(\boldsymbol{z}, D_{\boldsymbol{X}}, \boldsymbol{\mathcal{G}}) - \boldsymbol{F}(\boldsymbol{z})\|_2^2 \tag{4}$$

where $\boldsymbol{z} \sim \mathbb{P}_{\boldsymbol{X}}$ is chosen independent of the random dataset $D_{\boldsymbol{X}}$. To compute (4), we propose to
sample $n$ independent samples $\boldsymbol{X}_1', \ldots, \boldsymbol{X}_n'$ from $\mathbb{P}_{\boldsymbol{X}}$, leading to a new dataset $D_{\boldsymbol{X}'}$ independent of
$D_{\boldsymbol{X}}$, and we obtain the following optimzation problem:

$$\mathbb{E}_{\mathcal{S} \sim \mathbb{P}_{\mathcal{S}}} \|\mathcal{T}(D_{\boldsymbol{X}'}, D_{\boldsymbol{X}}, \boldsymbol{\mathcal{G}}) - \boldsymbol{F}(D_{\boldsymbol{X}'})\|_2^2 .$$

## 3.3 Inference with Cond-FiP

We provide a summary of inference procedure with Cond-FiP, with details in Appendix A.3.

**Observational Generation.** Cond-FiP is capable of generating new data samples: given a random
vector noise $\boldsymbol{n} \sim \mathbb{P}_{\boldsymbol{N}}$, we can estimate the observational sample associated according to an unknown
SCM $\mathcal{S}(\mathbb{P}_{\boldsymbol{N}}, \boldsymbol{\mathcal{G}}, \boldsymbol{F}) \sim \mathbb{P}_{\mathcal{S}}$ as long as we have access to its empirical representation $(D_{\boldsymbol{X}}, \boldsymbol{\mathcal{G}})$.
Formally, starting from $\boldsymbol{n}_0 = \boldsymbol{n}$, we infer the associated observation by computing for $\ell = 1, \ldots, d$:

$$\boldsymbol{n}_\ell = \mathcal{T}(\boldsymbol{n}_{\ell-1}, D_{\boldsymbol{X}}, \boldsymbol{\mathcal{G}}) + \boldsymbol{n} . \tag{5}$$

After (at most) $d$ iterations, $\boldsymbol{n}_d$ corresponds to the observational sample associated to the original
noise $\boldsymbol{n}$ according to our conditional SCM $\mathcal{T}(\cdot, D_{\boldsymbol{X}}, \boldsymbol{\mathcal{G}})$. To sample noise from $\mathbb{P}_{\boldsymbol{N}}$, we leverage
cond-FiP that can estimates noise samples under the ANM assumption by computing $\widetilde{D_{\boldsymbol{N}}} :=$
$D_{\boldsymbol{X}} - \mathcal{T}(D_{\boldsymbol{X}}, \mu(D_{\boldsymbol{X}}, \boldsymbol{\mathcal{G}}))$. From these estimated noise samples, we can efficiently estimate the joint
distribution of the noise by computing the inverse cdfs of the marginals as proposed in FiP.

**Interventional Generation.** Cond-FiP also enables the estimation of interventions given an empirical
representation $(D_{\boldsymbol{X}}, \boldsymbol{\mathcal{G}})$ of an unkown SCM $\mathcal{S}(\mathbb{P}_{\boldsymbol{N}}, \boldsymbol{\mathcal{G}}, \boldsymbol{F}) \sim \mathbb{P}_{\mathcal{S}}$. To achieve this, we start from a

noise sample $n$, and we generate the associated intervened sample $\widehat{z}^{\mathrm{do}}$ by directly modifying the conditional SCM provided by Cond-FiP. More specifically, we modify in place the SCM obtained by Cond-FiP, leading to its interventional version $\mathcal{T}^{\mathrm{do}}(\cdot, D_{\boldsymbol{X}}, \mathcal{G})$. Now, generating an intervened sample can be done by applying the loop defined in (5), starting from $n$ and using the intervened SCM $\mathcal{T}^{\mathrm{do}}(\cdot, D_{\boldsymbol{X}}, \mathcal{G})$ rather than the original one.

# 4 Experiments

We begin by describing our experimental setup in Section 4.1, and then present the results of our empirical analysis in Section 4.2, where we benchmark Cond-FiP against state-of-the-art baselines.

## 4.1 Setup

**Data Generation Process.** We use the synthetic data generation procedure proposed by Lorch et al. [2022] to generate SCMs as this framework supports a wide variety of SCMs, making it well-suited for amortized training. It allows sampling of graphs from different schemes and noise variables from diverse distributions. Further, we can also control the complexity of causal mechanisms, choosing between linear (*LIN*) functions or random fourier features (*RFF*) for non-linear causal mechanisms. We construct two distribution of SCMs, $\mathbb{P}_{\mathrm{IN}}$, and $\mathbb{P}_{\mathrm{OUT}}$, which vary based on the choice for sampling causal graphs, noise variables, and causal relationships, see Appendix B.1 for more details.

**Training Datasets.** We randomly sample $\simeq 4e6$ SCMs from the $\mathbb{P}_{\mathrm{IN}}$ distribution, each with $d = 20$ total nodes. From each SCM, we extract the causal graph $\mathcal{G}$ and generate $n_{\mathrm{train}} = 400$ observations to obtain $D_{\boldsymbol{X}}$. This procedure is used to generate training data both the dataset encoder and Cond-FiP, with each epoch containing $\simeq 400$ randomly generated datasets.

**Test Datasets.** We evaluate the model's generalization both in-distribution and out-of-distribution by sampling test datasets from $\mathbb{P}_{\mathrm{IN}}$ and $\mathbb{P}_{\mathrm{OUT}}$, respectively. The test datasets are categorized as follows: LIN **IN** and RFF **IN** where the SCM are sampled from $\mathbb{P}_{\mathrm{IN}}$ with linear and non-linear causal mechanisms respectively. Similarly, we define LIN **OUT** and RFF **OUT** where the SCMs are sampled from $\mathbb{P}_{\mathrm{OUT}}$ instead.

For each category, we vary the total nodes $d \in [10, 20, 50, 100]$ and sample 6 or 9 SCMs per $d$, based on the available schemes for sampling the causal graphs (check Appendix B.1 for details). This results in a total of 120 test datasets, supporting a comprehensive evaluation of the methods. For each SCM we generate $n_{\mathrm{test}} = 800$ samples, split equally into task context $D_{\boldsymbol{X}}$ and queries $D_{\boldsymbol{X}'}$ for evaluation. An interesting aspect of our test setup is we assess the model's ability to generalize to larger graphs ($d = 50$, $d = 100$), despite training only with $d = 20$ node graphs.

**Model Architecture.** For both the dataset encoder and cond-FiP, we set the embedding dimension to $d_h = 256$ and the hidden dimension of MLP blocks to 512. Both of our transformer-based models contains 4 attention layers and each attention consists of 8 attention heads. Please check Appendix B.2 for further details and Cond-FiP's memory and compute requirements.

**Baselines.** We compare Cond-FiP against FiP [Scetbon et al., 2024], DECI [Geffner et al., 2022], and DoWhy [Blöbaum et al., 2022]. Since the baselines do not have any amortization procedure, they are trained from scratch on each test setting. For a fair comparison with our method, we use the same context set $D_{\boldsymbol{X}}$ with 400 samples to train the baselines, which was used to obtain the dataset embeddings in Cond-FiP. All the methods are then evaluated on the remaining 400 samples in query set $D_{\boldsymbol{X}'}$. Also, we provide the true graph $\mathcal{G}$ to all the baselines to ensure consistency with Cond-FiP.

To avoid potential confusion, we clarify that the notion of distribution shift is defined w.r.t Cond-FiP's training setup. For the baselines, there is no distribution shift as they are trained on the context ($D_{\boldsymbol{X}}$) drawn from the specific test distribution. The most important comparison is with the baseline FiP, as Cond-FiP is its amortized counterpart. Further, we do not report detailed comparisons with CausalNF [Javaloy et al., 2023] as its performance was consistently weaker than other baselines, check Appendix J for details.

**Evaluation Tasks.** We evaluate the methods on the following three tasks. *Noise Prediction:* given the observations $D_{\boldsymbol{X}}$ and the true graph $\mathcal{G}$, infer the noise variables $\widehat{D_{\boldsymbol{N}}}$. *Sample Generation:* given the noise samples $D_{\boldsymbol{N}}$ and the true graph $\mathcal{G}$, generate the causal variables $\widehat{D_{\boldsymbol{X}}}$. *Interventional Generation:* generate intervened samples from noise samples $D_{\boldsymbol{N}}$ and the true graph $\mathcal{G}$.

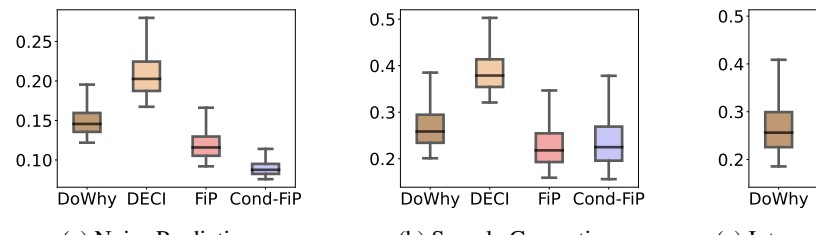

(a) Noise Prediction    (b) Sample Generation    (c) Interventional Generation

Figure 2: **In-Distribution Results.** Benchmarking Cond-FiP for various evaluation tasks, with datasets sampled from RFF **IN** with $d = 20$. The y-axis denotes the RMSE, with mean and standard error over the respective test datasets. Results indicate Cond-FiP can generalize to novel in-distribution instances, with detailed results in Appendix C.

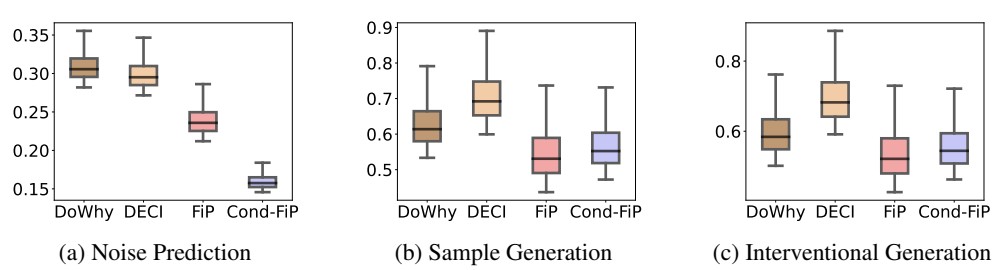

(a) Noise Prediction    (b) Sample Generation    (c) Interventional Generation

Figure 3: **OOD Results.** Benchmarking Cond-FiP for various evaluation tasks, with datasets sampled from RFF **OUT** with $d = 100$ to test for OOD generalization. The y-axis denotes the RMSE, with mean and standard error over the respective test datasets. Results indicate Cond-FiP can generalize to novel OOD instances and larger graphs, with detailed results in Appendix C.

**Metric.** Let us denote a predicted & true target as $\widehat{Y} \in \mathbb{R}^{n_{\text{test}} \times d}$ and $Y \in \mathbb{R}^{n_{\text{test}} \times d}$. Then RMSE is computed as $\frac{1}{n_{\text{test}}} \sum_{i=1}^{n_{\text{test}}} \sqrt{\frac{1}{d} \|[Y]_i - [\widehat{Y}]_i\|_2^2}$. Note that we scale RMSE by dimension $d$, which allows us to compare results across different graph sizes.

## 4.2 Results

**Generalization to OOD data and larger graphs.** In Figure 2, we first present results for in-distribution generalization using test datasets sampled from RFF **IN** for graphs with $d = 20$ nodes. Cond-FiP performs competitively with baselines trained from scratch on each test instance, hence it successfully generalizes to novel in-distribution instances. Notably, Cond-FiP was never explicitly trained to generate interventional data, and its strong performance on this task further supports that it captures the underlying causal mechanisms.

Next we consider the more challenging case of OOD generalization using test datasets sampled from RFF **OUT** and graphs with $d = 100$ nodes, while the Cond-FiP was trained only with $d = 20$ node graphs. As shown in Figure 3, Cond-FiP continues to perform well, indicating successful generalization to OOD instances and significantly larger graphs! Due to space constraints, we report results for SCMs with non-linear mechanisms—the more challenging setting. Full results for both in-distribution and OOD scenarios are available in Appendix C, where our findings remain consistent.

We also assess Cond-FiP's sensitivity to distribution shifts by varying the magnitude of distribution shift (details in Appendix D). We consider two cases, where we control the severity in distribution shift by controlling the causal mechanisms or the noise variables. We find that Cond-FiP is more robust to shifts in causal mechanisms, with minimal performance degradation. However, its performance is more sensitive to shifts in noise distributions, deteriorating as the magnitude of shift increases.

**Better Generalization in Scare Data Regimes.** An advantage of amortized inference methods is their ability to generalize well when context $D_X$ for test instances is small. As the context size decreases, baselines often suffer significant performance drops as they require training from scratch. In contrast, Cond-FiP is less impacted as its parameters remain unchanged at inference time, and

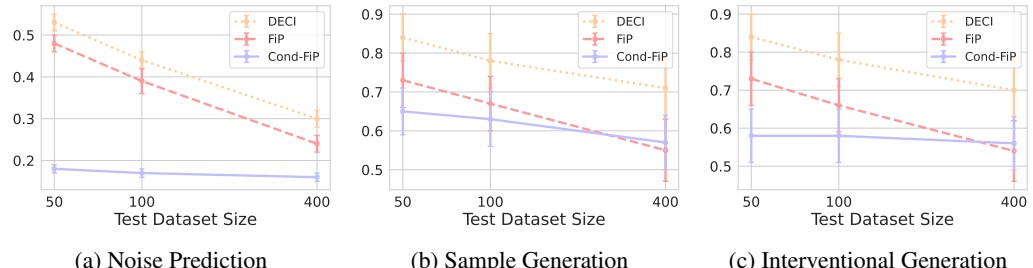

(a) Noise Prediction      (b) Sample Generation      (c) Interventional Generation

Figure 4: **Scarce Data Regime Results.** Benchmarking Cond-FiP on the various evaluation tasks (RFF **OUT** and $d = 100$) as we reduce the test dataset size. The y-axis denotes the RMSE, with mean and standard error over the respective test datasets. Cond-FiP generalizes much better than the baselines in the low-data regime, with detailed results in Appendix E.

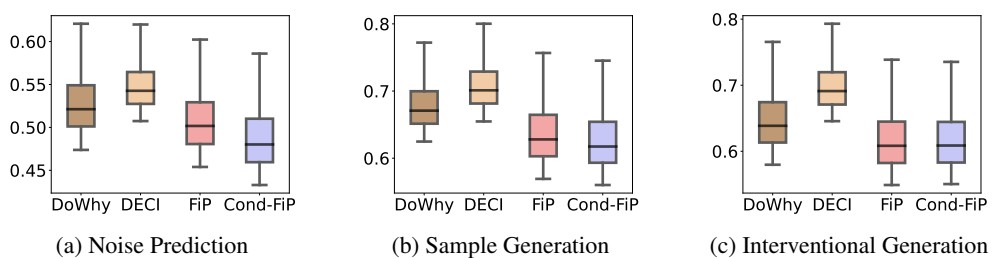

(a) Noise Prediction      (b) Sample Generation      (c) Interventional Generation

Figure 5: **OOD Results without True Graph.** Benchmarking Cond-FiP for various evaluation tasks, with datasets sampled from RFF **OUT** with $d = 100$ where the true graph $\mathcal{G}$ is not present in input context, rather its inferred via AVICI. The y-axis denotes the RMSE, with mean and standard error over the respective test datasets. Results indicate Cond-FiP can generalize to novel instances even in the absence of true graph, with detailed results in Appendix F.

the inductive bias learned during training enables effective generalization even with limited context. In Figure 4, we demonstrate this in the challenging OOD setting (RFF **OUT**;, $d = 100$), where Cond-FiP outperforms the baselines. Please check Appendix E for further details.

**Generalization without True Causal Graph.** So far, our results assume access to the true causal graph ($\mathcal{G}$) as part of the input context to Cond-FiP. However, Cond-FiP can be extended to operate without this information by first inferring the graph using amortized structure learning methods [Lorch et al., 2022, Ke et al., 2022]. We demonstrate this in Figure 5 for the RFF **OUT**; setting with $d = 100$ nodes, using graphs inferred via AVICI [Lorch et al., 2022] for both Cond-FiP and the baselines. The results show that Cond-FiP remains competitive, further supporting its ability to capture underlying causal mechanisms. Please check Appendix F for more details.

**Ablation Study.** We conduct ablation studies on both the encoder and decoder to better understand how the training data affects generalization performance. We find that Cond-FiP remains competitive even when the encoder is trained on only RFF data, compared to training on a mixture of both. In contrast, decoder performance benefits more noticeably from training on the combined dataset. Please check Appendix G.1 and G.2 for more details regarding the ablation experiments.

**Generalization to novel data simulators.** We further evaluate Cond-FiP on test datasets generated using C-Suite [Geffner et al., 2022], a synthetic data simulator distinct from the training simulator. As shown in Figure 6, Cond-FiP generalizes well to these novel instances. Additionally, we consider a modified C-Suite benchmark with Gaussian mixture model noise. Results in Figure 7, Appendix H show that Cond-FiP also generalizes to instances with more complex noise distributions.

Finally, we show that Cond-FiP can generalize to the real-world instances using the Sachs dataset [Sachs et al., 2005]. Although Cond-FiP cannot be trained on real-world datasets since the encoder requires access to true noise variables, it can still be used for inference. We evaluate the quality of generated samples by comparing them to observed data using the Maximum Mean Discrepancy (MMD) metric [Gretton et al., 2012]. See Appendix I for more details.

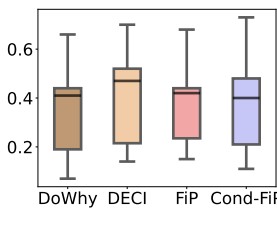 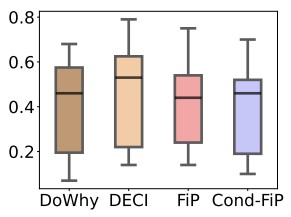 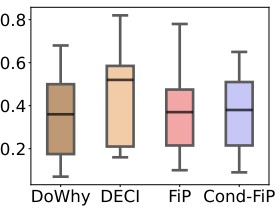

(a) Noise Prediction     (b) Sample Generation     (c) Interventional Generation

Figure 6: **CSuite Results.** Benchmarking Cond-FiP on the various evaluation tasks on the CSuite benchmark, which uses a different data simulator than the Cond-FiP's training data simulator. The y-axis denotes the RMSE, with mean and standard error across the 9 test datasets.

## 5 Related Works

**Amortized Causal Learning.** Amortized inference has gained traction in causality research, particularly for structure learning. Early works by Lorch et al. [2022] and Ke et al. [2022] introduced transformer-based models trained on multiple synthetic datasets using supervised objectives for amortized inference of causal structure. Their approach aligns with recent works on in-context learning of function classes via transformers [Müller et al., 2021, Akyürek et al., 2022, Garg et al., 2022, Von Oswald et al., 2023]. Subsequent improvements targeted OOD generalization [Wu et al., 2024] and applications to gene regulatory networks [Ke et al., 2023]. Beyond structure learning, amortized methods have been developed for ATE estimation [Nilforoshan et al., 2023, Zhang et al., 2023, Sauter et al., 2025], model selection [Gupta et al., 2023], and partial causal discovery tasks such as learning topological order [Scetbon et al., 2024]. However, amortized inference of causal mechanisms in SCMs remains unaddressed, which is the central focus of our work.

**Autoregressive Causal Learning.** Most causal discovery methods focus first on structure learning [Chickering, 2002, Peters et al., 2014, Zheng et al., 2018], followed by per-node maximum likelihood estimation to recover the causal mechanisms [Blöbaum et al., 2022]. In contrast, recent works on causal autoregressive flows [Khemakhem et al., 2021, Geffner et al., 2022, Javaloy et al., 2023] focus on SOTA normalizing flow based generative models to infer causal mechanisms. Further, FiP [Scetbon et al., 2024] modeled SCMs as fixed-point problems over causal (topological) ordering of nodes using transformer-based architectures. These approaches efficiently learn SCMs but require training a separate model per dataset. In this work, we extend FiP to enable amortized inference of causal mechanisms across different SCM instances, removing this limitation.

## 6 Conclusion

In this work, we propose novel methodology for training a *single* model for amortized inference of SCMs. Cond-FiP not only generalizes to unseens in-distribution instances, but also to a wide range of OOD instances, including larger graphs, complex noise distributions, and real-world data. To the best of our knowledge, this is the first approach to demonstrate the feasibility of learning causal mechanisms in a reusable, foundational manner—paving the way for a paradigmatic shift towards the assimilation of causal knowledge across datasets.

**Limitations.** Our training is limited to synthetic additive noise SCMs due to the requirement of true noise variables for learning the dataset encoder. However, the conditional FiP decoder (see Section 3.2) does not rely on this assumption and can be applied to general SCMs given pretrained dataset embeddings. A promising direction for future work is to explore more general encoding schemes, such as self-supervised learning, or design an implicit in-context learning approach to remove the need for dataset embeddings via direct attention over the context [Mittal et al., 2024].

While Cond-FiP generalizes to larger graphs, it does not yet benefit from larger context sizes at inference (Appendix K.1), suggesting the need to scale both the model and training data for richer contexts. Additionally, although Cond-FiP performs well on generating interventional samples, it doesn't perform well on counterfactual generation (Appendix K.2). Future work will explore scaling Cond-FiP to larger problem instances and application for more complex tasks (counterfactual generation) in real-world scenarios.

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

# Appendix

## Table of Contents

# A  Additional Details on Cond-FiP

## A.1  DAG-Attention Mechanism

In FiP [Scetbon et al., 2024] the authors propose to leverage the transformer architecture to learn SCMs from observations. By reparameterizing an SCM according to a topological ordering induced by its graph, the authors show that any SCM can be reformulated as a fixed-point problem of the form $\boldsymbol{X} = \boldsymbol{H}(\boldsymbol{X}, \boldsymbol{N})$ where $\boldsymbol{H}$ admits a simple triangular structure:

$$[\mathrm{Jac}_{\boldsymbol{x}}\boldsymbol{H}(\boldsymbol{x}, \boldsymbol{n})]_{i,j} = 0, \quad \text{if} \quad j \geq i$$
$$[\mathrm{Jac}_{\boldsymbol{n}}\boldsymbol{H}(\boldsymbol{x}, \boldsymbol{n})]_{i,j} = 0, \quad \text{if} \quad i \neq j,$$

where $\mathrm{Jac}_{\boldsymbol{x}}\boldsymbol{H}$, $\mathrm{Jac}_{\boldsymbol{n}}\boldsymbol{H}$ denote the Jacobian of $\boldsymbol{H}$ w.r.t the first and second variables respectively. Motivated by this fixed-point reformulation, FiP considers a transformer-based architecture to model the functional relationships of SCMs and propose a new attention mechanism to represent DAGs in a differentiable manner. Recall that the standard attention matrix is defined as:

$$\mathrm{A}_M(\boldsymbol{Q}, \boldsymbol{K}) = \frac{\exp((\boldsymbol{Q}\boldsymbol{K}^T - \boldsymbol{M})/\sqrt{d_h})}{\exp((\boldsymbol{Q}\boldsymbol{K}^T - \boldsymbol{M})/\sqrt{d_h})\,\mathbf{1}_d} \tag{6}$$

where $\boldsymbol{Q}, \boldsymbol{K} \in \mathbb{R}^{d \times d_h}$ denote the keys and queries for a single attention head, and $\boldsymbol{M} \in \{0, +\infty\}^{d \times d}$ is a (potential) mask. When M is chosen to be a triangular mask, the attention mechanism (6) enables to parameterize the effects of previous nodes on the current one However, the normalization inherent to the softmax operator in standard attention mechanisms prevents effective modeling of root nodes, which *should not* be influenced by any other node in the graph. To alleviate this issue, FiP proposes to consider the following formulation instead:

$$\mathrm{DA}_M(\boldsymbol{Q}, \boldsymbol{K}) = \frac{\exp((\boldsymbol{Q}\boldsymbol{K}^T - \boldsymbol{M})/\sqrt{d_h})}{\mathcal{V}\big(\,\exp((\boldsymbol{Q}\boldsymbol{K}^T - \boldsymbol{M})/\sqrt{d_h})\,\mathbf{1}_d\big)} \tag{7}$$

where $\mathcal{V}_i(\boldsymbol{v}) = v_i$ if $v_i \geq 1$, else $\mathcal{V}_i(\boldsymbol{v}) = 1$ for any $\boldsymbol{v} \in \mathbb{R}^d$. While softmax forces the coefficients along each row of the attention matrix to sum to one, the attention mechanism described in (7) allows the rows to sum in $[0, 1]$, thus enabling to model root nodes in attention.

## A.2  Details on Encoder Training

**Additive Noise Model Assumption.**  Our method relies on the ANM assumption only for the training the encoder. This is because we require the encoder to predict the noise from data in order to obtain embeddings, and under the ANM assumption, the mapping from data to noise can be easily expressed as $x \rightarrow x - F(x)$ where $F$ is the generative functional mechanism of the generative ANM. However, if we were to consider general SCMs, i.e. of the form $X = F(X, N)$, we would need access to the mapping $x \rightarrow F^{-1}(x, \cdot)(x)$ (assuming this function is invertible), which for general functions is not tractable. Also, note that the ANM assumption by default ensures invertibility since the jacobian w.r.t noise is a triangular matrix with nonzero diagonal elements. An interesting future work would be to consider a more general dataset encoding (using self-supervised techniques) that do not require the ANM assumption, but we believe this is out of the scope of this work.

We now provide further details on training the encoder and show how recovering the noise is equivalent to learn the inverse causal generative process. Recall that an SCM is an *implicit* generative model that, given a noise sample **N**, generates the corresponding observation according to the following fixed-point equation in **X**

$$\mathbf{X} = F(\mathbf{X}, \mathbf{N})$$

More precisely, to generate the associated observation, one must solve the above fixed-point equation in **X** given the noise **N**. Let us now introduce the following notation that will be instrumental for the subsequent discussion: we denote $F_{\mathbf{N}}(z) : z \rightarrow F(z, \mathbf{N})$.

Due to the specific structure of $F$ (determined by the DAG $\mathcal{G}$ associated with the SCM), the fixed-point equation mentioned above can be efficiently solved by iteratively applying the function $F_{\mathbf{N}}$

to the noise (see Eq. (5) in the manuscript). As a direct consequence, the observation $\mathbf{X}$ can be expressed as a function of the noise:

$$\mathbf{X} = F_{\text{gen}}(\mathbf{N})$$

where $F_{\text{gen}}(\mathbf{N}) := (F_{\mathbf{N}})^{\circ d}(\mathbf{N})$, $d$ is the number of nodes, and $\circ$ denotes the composition operation. In the following we refer to $F_{\text{gen}}$ as the *explicit* generative model induced by the SCM.

Conversely, assuming that the mapping $z \to F_{\text{gen}}(z)$ is invertible, then one can express the noise as a function of the data:

$$\mathbf{N} = F_{\text{gen}}^{-1}(\mathbf{X})$$

Therefore, learning to recover the noise from observation is equivalent to learn the function $F_{\text{gen}}^{-1}$, which is exactly the inverse of the explicit generative model $F_{\text{gen}}$. It is also worth noting that under the ANM assumption (i.e. $F(\mathbf{X}, \mathbf{N}) = f(\mathbf{X}) + \mathbf{N}$), $F_{\text{gen}}$ is in fact always invertible and its inverse admits a simple expression which is

$$F_{\text{gen}}^{-1}(z) = z - f(z)$$

Therefore, in this specific case, learning the inverse generative model $F_{\text{gen}}^{-1}$ is exactly equivalent to learning the causal mechanism function $f$.

### A.3 Inference with Cond-FiP

**Sample Generation.** Given a dataset $D_{\boldsymbol{X}}$ and its causal graph $\mathcal{G}$, we denote $z \to \mathcal{T}(z, D_{\boldsymbol{X}}, \mathcal{G})$ the function infered by Cond-FiP. This function defines the predicted SCM obtained by our model, and we can directly use it to generate new points. More precisely, given a noise sample $\mathbf{n}$, we can generate the associated observational sample by solving the following equation in $\mathbf{x}$:

$$\mathbf{x} = \mathcal{T}(\mathbf{x}, D_{\boldsymbol{X}}, \mathcal{G}) + \mathbf{n}$$

To solve this fixed-point equation, we rely on the fact that $\mathcal{G}$ is a DAG, which enables to solve the fixed-point problem using the following simple iterative procedure. Starting with $\boldsymbol{z}_0 = \mathbf{n}$, we compute for $\ell = 1, \ldots, d$ where $d$ is the number of nodes

$$\boldsymbol{z}_\ell = \mathcal{T}(\boldsymbol{z}_{\ell-1}, D_{\boldsymbol{X}}, \mathcal{G}) + \mathbf{n}$$

After $d$ iterations we obtain the following,

$$\boldsymbol{z}_d = \mathcal{T}(\boldsymbol{z}_d, D_{\boldsymbol{X}}, \mathcal{G}) + \mathbf{n}$$

Therefore, $\boldsymbol{z}_d$ is the solution of the fixed-point problem above, which corresponds to the observational sample associated to $\mathbf{n}$ according to our predicted SCM $z \to \mathcal{T}(z, D_{\boldsymbol{X}}, \mathcal{G})$.

**Interventional Prediction.** Recall that given a dataset $D_{\boldsymbol{X}}$ and its causal graph $\mathcal{G}$, $z \in \mathbb{R}^d \to \mathcal{T}(z, D_{\boldsymbol{X}}, \mathcal{G}) \in \mathbb{R}^d$ denotes the SCM infered by Cond-FiP. Let us also denote the coordinate-wise formulation of our SCM defined for any $z \in \mathbb{R}^d$ as $\mathcal{T}(z, D_{\boldsymbol{X}}, \mathcal{G}) = [[\mathcal{T}(z, D_{\boldsymbol{X}}, \mathcal{G})]_1, \ldots, [\mathcal{T}(z, D_{\boldsymbol{X}}, \mathcal{G})]_d]$, where for all $i \in \{1, \ldots, d\}$, $z \in \mathbb{R}^d \to [\mathcal{T}(z, D_{\boldsymbol{X}}, \mathcal{G})]_i \in \mathbb{R}$ is a real-valued function.

In order to intervene on this predicted SCM, we simply have to modify in place the predicted function. For example, assume that we want to perform the following intervention $\text{do}(X_i) = a$. Then, to obtain the intervened SCM, we define a new function $z \to \mathcal{T}^{\text{do}(X_i)=a}(z, D_{\boldsymbol{X}}, \mathcal{G})$ defined for any $z \in \mathbb{R}^d$ as: $[\mathcal{T}^{\text{do}(X_i)=a}(z, D_{\boldsymbol{X}}, \mathcal{G})]_j := [\mathcal{T}(z, D_{\boldsymbol{X}}, \mathcal{G})]_j$ if $j \neq i$ and $[\mathcal{T}^{\text{do}(X_i)=a}(z, D_{\boldsymbol{X}}, \mathcal{G})]_i := a$.

Now, using this intervened SCM $z \to \mathcal{T}^{\text{do}(X_i)=a}(z, D_{\boldsymbol{X}}, \mathcal{G})$, we can apply the exact same generation procedure as the one introduced above to generate intervened samples according to our intervened SCM.

## B Details on Experiment Setup with AVICI Benchmark

### B.1 AVICI Benchmark

We use the synthetic data generation procedure proposed by Lorch et al. [2022] to generate SCMs in our empirical study. It provides access to a wide variety of SCMs, hence making it an excellent setting for amortized training.

- **Graphs:** We have the option to sample graphs as per the following schemes: Erods-Renyi [Erdos and Renyi, 1959], scale-free models [Barabási and Albert, 1999], Watts-Strogatz [Watts and Strogatz, 1998], and stochastic block models [Holland et al., 1983].
- **Noise Variables:** To sample noise variables, we can choose from either the gaussian or laplace distribution where variances are sampled randomly.
- **Functional Mechanisms:** We can control the complexity of causal relationships: either we set them to be linear (LIN) functions randomly sampled, or use random fourier features (RFF) for generating random non-linear causal relationships.

We construct two distribution of SCMs $\mathbb{P}_{IN}$, and $\mathbb{P}_{OUT}$, which vary based on the choice for sampling causal graphs, noise variables, and causal relationships. The classification aids in understanding the creation of train and test datasets.

- **In-Distribution ($\mathbb{P}_{IN}$):** We sample causal graphs using the Erods-Renyi and scale-free models schemes. Noise variables are sampled from the gaussian distribution, and we allow for both LIN and RFF causal relationships.
- **Out-of-Distribution ($\mathbb{P}_{OUT}$):** Causal graphs are drawn from Watts-Strogatz and stochastic block models schemes. Noise variables follow the laplace distribution, and both the LIN and RFF cases are used to sample functions. However, the parameters of these distributions are sampled from a different range as compared to $\mathbb{P}_{IN}$ to create a distribution shift.

We provide further details on the shift in the support of parameters for functional mechanisms below. For complete details please refer to Table 3, Appendix in Lorch et al. [2022].

- **Linear Functional Mechanism.**
  - *In-Distribution ($\mathbb{P}_{IN}$)*
    * Weights: $\sim U_{\pm}(1,3)$, Bias $\sim U(-3,3)$.
  - *Out-of-Distribution ($\mathbb{P}_{OUT}$)*
    * Weights: $\sim U_{\pm}(0.5,2) \cup U_{\pm}(2,4)$, Bias $\sim U(-3,3)$.
- **RFF Functional Mechanism.**
  - *In-Distribution ($\mathbb{P}_{IN}$)*
    * Length Scale: $\sim U(7,10)$, Output Scale: $\sim U(5,8) \cup U(8,12)$, Bias $\sim U_{\pm}(-3,3)$.
  - *Out-of-Distribution ($\mathbb{P}_{OUT}$):*
    * Length Scale: $\sim U(10,20)$, Output Scale: $\sim U(8,12) \cup U(18,22)$, Bias $\sim U_{\pm}(-3,3)$.

**Test Datasets.**

- LIN **IN**: SCMs sampled from $\mathbb{P}_{IN}$ with linear causal mechanisms. We have 3 different options for sampling graphs in this case, and we randomly sample 3 different SCMs for each scenario, leading to a total of 9 instances.
- RFF **IN**: SCMs sampled from $\mathbb{P}_{IN}$ with non-linear causal mechanisms. We have 3 different options for sampling graphs in this case, and we randomly sample 3 different SCMs for each scenario, leading to a total of 9 instances.
- LIN **OUT**: SCMs sampled from $\mathbb{P}_{OUT}$ with linear causal mechanisms. We have 2 different options for sampling graphs in this case, and we randomly sample 3 different SCMs for each scenario, leading to a total of 6 instances.
- RFF **OUT**: SCMs sampled from $\mathbb{P}_{OUT}$ with non-linear causal mechanisms. We have 2 different options for sampling graphs in this case, and we randomly sample 3 different SCMs for each scenario, leading to a total of 6 instances.

## B.2 Model Architecture and Training Details

For both the dataset encoder and cond-FiP, we set the embedding dimension to $d_h = 256$ and the hidden dimension of MLP blocks to $512$. Both of our transformer-based models contains 4 attention layers and each attention consists of 8 attention heads. The models were trained for a total of $10k$ epochs with the Adam optimizer [Paszke et al., 2017], where we used a learning rate of $1e-4$ and a weight decay of $5e-9$. Each epoch contains $\simeq 400$ randomly generated datasets from the distribution $\mathbb{P}_{\text{IN}}$. We also use the EMA implementation of [Karras et al., 2023] to train our models.

**Memory Requirements.**   We trained Cond-FiP on a single L40 GPU with 48GB of memory, using an effective batch size of $8$ with gradient accumulation. We outline the detailed memory computation as follows:

- Each batch consists of $n = 400$ samples with dimension $d = 20$ requiring less than $1$ MiB of data in FP32 precision.
- Storing the model on the GPU requires under 100 MiB.
- Our transformer architecture has $4$ attention layers, a 256-dimensional embedding space, and a 512-dimensional feedforward network. Using a standard (non-flash) attention implementation, a forward pass consumes approximately 30 GiB of GPU memory.

Compared to the baselines, Cond-FiP has similar memory requirements to DECI [Geffner et al., 2022] and FiP [Scetbon et al., 2024], as all three train neural networks of comparable size. The main exception is DoWhy [Blöbaum et al., 2022], which fits simpler models for each node, but this approach does not scale well as the graph size increases.

**Computational Cost.**   Like other amortized approaches, Cond-FiP has a higher training cost than the baselines, as it is trained across multiple datasets. While the cost of each forward-pass is comparable to FiP, we trained Cond-FiP over approximately 4M datasets in an amortized manner. However, Cond-FiP offers a significant advantage at inference time since it requires only a single forward pass to generate predictions, whereas the baselines must be retrained from scratch for each new dataset. Thus, while Cond-FiP incurs a higher one-time training cost, its substantially faster at inference.

## B.3 Code Repository

We plan to open-source the code along with comprehensive documentation to facilitate reproducibility of our experiments. For the submission, we have prepared an anonymized version of the codebase, which can be accessed via this link: `https://anonymous.4open.science/r/neurips_2025_condfip-1277/`.

Please note that while the codebase is not directly executable, it provides full access to the implementation of all components of our framework:

- *cond-fip/models* contains the implementation of the transformer-based encoder and the Cond-FIP architecture.
- *cond-fip/tasks* includes the training and inference methods associated with our framework.

 # C    Complete Results for Cond-FiP on AVICI Benchmark

| Method | Total Nodes | LIN **IN** | RFF **IN** | LIN **OUT** | RFF **OUT** |
|---|---|---|---|---|---|
| DoWhy | 10 | 0.03 (0.0) | 0.13 (0.02) | 0.04 (0.01) | 0.11 (0.01) |
| DECI | 10 | 0.09 (0.01) | 0.23 (0.03) | 0.12 (0.01) | 0.23 (0.03) |
| FiP | 10 | 0.04 (0.0) | 0.09 (0.01) | 0.06 (0.01) | 0.08 (0.01) |
| Cond-FiP | 10 | 0.06 (0.01) | 0.10 (0.01) | 0.07 (0.01) | 0.10 (0.01) |
| DoWhy | 20 | 0.03 (0.01) | 0.15 (0.02) | 0.03 (0.0) | 0.23 (0.01) |
| DECI | 20 | 0.10 (0.02) | 0.21 (0.03) | 0.08 (0.02) | 0.23 (0.02) |
| FiP | 20 | 0.04 (0.0) | 0.12 (0.02) | 0.05 (0.0) | 0.15 (0.02) |
| Cond-FiP | 20 | 0.06 (0.01) | 0.09 (0.01) | 0.07 (0.0) | 0.12 (0.0) |
| DoWhy | 50 | 0.03 (0.0) | 0.18 (0.03) | 0.03 (0.0) | 0.29 (0.03) |
| DECI | 50 | 0.09 (0.01) | 0.24 (0.02) | 0.07 (0.01) | 0.29 (0.02) |
| FiP | 50 | 0.04 (0.0) | 0.14 (0.03) | 0.04 (0.0) | 0.23 (0.04) |
| Cond-FiP | 50 | 0.06 (0.01) | 0.10 (0.01) | 0.07 (0.01) | 0.14 (0.01) |
| DoWhy | 100 | 0.03 (0.0) | 0.20 (0.03) | 0.03 (0.0) | 0.31 (0.02) |
| DECI | 100 | 0.08 (0.02) | 0.26 (0.03) | 0.07 (0.01) | 0.30 (0.02) |
| FiP | 100 | 0.04 (0.0) | 0.16 (0.03) | 0.04 (0.0) | 0.24 (0.02) |
| Cond-FiP | 100 | 0.05 (0.0) | 0.10 (0.01) | 0.07 (0.01) | 0.16 (0.01) |

Table 1: **Results for Noise Prediction.** We compare Cond-FiP against the baselines for the task of predicting noise variables from the input observations. Each cell reports the mean (standard error) RMSE over the multiple test datasets for each scenario. Shaded rows denote the case where the graph size is larger than the train graph sizes ($d = 20$) for Cond-FiP. *Results show that Cond-FiP generalizes to both in-distribution and OOD instances.*

| Method | Total Nodes | LIN **IN** | RFF **IN** | LIN **OUT** | RFF **OUT** |
|---|---|---|---|---|---|
| DoWhy | 10 | 0.05 (0.0) | 0.18 (0.03) | 0.06 (0.01) | 0.12 (0.02) |
| DECI | 10 | 0.15 (0.02) | 0.33 (0.04) | 0.16 (0.02) | 0.27 (0.03) |
| FiP | 10 | 0.07 (0.0) | 0.13 (0.02) | 0.08 (0.01) | 0.11 (0.02) |
| Cond-FiP | 10 | 0.06 (0.01) | 0.14 (0.02) | 0.05 (0.01) | 0.08 (0.01) |
| DoWhy | 20 | 0.06 (0.01) | 0.27 (0.05) | 0.05 (0.0) | 0.39 (0.04) |
| DECI | 20 | 0.16 (0.02) | 0.39 (0.05) | 0.13 (0.02) | 0.44 (0.04) |
| FiP | 20 | 0.08 (0.01) | 0.23 (0.05) | 0.08 (0.01) | 0.27 (0.04) |
| Cond-FiP | 20 | 0.05 (0.01) | 0.24 (0.06) | 0.07 (0.01) | 0.30 (0.03) |
| DoWhy | 50 | 0.08 (0.01) | 0.35 (0.09) | 0.06 (0.01) | 0.54 (0.06) |
| DECI | 50 | 0.15 (0.01) | 0.46 (0.06) | 0.13 (0.02) | 0.67 (0.06) |
| FiP | 50 | 0.09 (0.01) | 0.26 (0.05) | 0.08 (0.01) | 0.48 (0.06) |
| Cond-FiP | 50 | 0.08 (0.01) | 0.25 (0.05) | 0.07 (0.0) | 0.48 (0.07) |
| DoWhy | 100 | 0.06 (0.0) | 0.33 (0.07) | 0.06 (0.01) | 0.63 (0.07) |
| DECI | 100 | 0.14 (0.02) | 0.50 (0.09) | 0.14 (0.02) | 0.71 (0.08) |
| FiP | 100 | 0.08 (0.01) | 0.3 (0.06) | 0.09 (0.01) | 0.55 (0.08) |
| Cond-FiP | 100 | 0.07 (0.01) | 0.29 (0.07) | 0.09 (0.01) | 0.57 (0.07) |

Table 2: **Results for Sample Generation.** We compare Cond-FiP against the baselines for the task of generating samples from the input noise variables. Each cell reports the mean (standard error) RMSE over the multiple test datasets for each scenario. Shaded rows denote the case where the graph size is larger than the train graph sizes ($d = 20$) for Cond-FiP. *Results show that Cond-FiP generalizes to both in-distribution and OOD instances.*

| Method | Total Nodes | LIN **IN** | RFF **IN** | LIN **OUT** | RFF **OUT** |
|---|---|---|---|---|---|
| DoWhy | 10 | 0.08 (0.03) | 0.19 (0.04) | 0.05 (0.01) | 0.12 (0.02) |
| DECI | 10 | 0.17 (0.02) | 0.34 (0.04) | 0.13 (0.02) | 0.25 (0.03) |
| FiP | 10 | 0.08 (0.01) | 0.15 (0.02) | 0.07 (0.01) | 0.09 (0.01) |
| Cond-FiP | 10 | 0.10 (0.03) | 0.21 (0.03) | 0.07 (0.01) | 0.11 (0.01) |
| DoWhy | 20 | 0.06 (0.01) | 0.27 (0.06) | 0.05 (0.0) | 0.36 (0.03) |
| DECI | 20 | 0.16 (0.02) | 0.38 (0.05) | 0.15 (0.04) | 0.42 (0.03) |
| FiP | 20 | 0.09 (0.01) | 0.23 (0.05) | 0.12 (0.04) | 0.25 (0.03) |
| Cond-FiP | 20 | 0.09 (0.01) | 0.24 (0.05) | 0.14 (0.03) | 0.31 (0.03) |
| DoWhy | 50 | 0.08 (0.01) | 0.29 (0.05) | 0.06 (0.01) | 0.53 (0.06) |
| DECI | 50 | 0.17 (0.02) | 0.44 (0.06) | 0.13 (0.02) | 0.64 (0.06) |
| FiP | 50 | 0.11 (0.02) | 0.25 (0.05) | 0.09 (0.01) | 0.46 (0.06) |
| Cond-FiP | 50 | 0.13 (0.02) | 0.27 (0.04) | 0.12 (0.02) | 0.48 (0.07) |
| DoWhy | 100 | 0.05 (0.0) | 0.33 (0.07) | 0.06 (0.01) | 0.60 (0.07) |
| DECI | 100 | 0.14 (0.02) | 0.49 (0.08) | 0.15 (0.02) | 0.70 (0.08) |
| FiP | 100 | 0.08 (0.01) | 0.29 (0.07) | 0.10 (0.01) | 0.54 (0.08) |
| Cond-FiP | 100 | 0.10 (0.01) | 0.30 (0.06) | 0.14 (0.02) | 0.56 (0.07) |

Table 3: **Results for Interventional Generation.** We compare Cond-FiP against the baselines for the task of generating interventional data from the input noise variables. Each cell reports the mean (standard error) RMSE over the multiple test datasets for each scenario. Shaded rows denote the case where the graph size is larger than the train graph sizes ($d = 20$) for Cond-FiP. *Results show that Cond-FiP generalizes to both in-distribution and OOD instances.*

# D    Experiments on Sensitivity to Distribution Shifts on AVICI benchmark

In Appendix C (Table 1, Table 2, Table 3), we tested OOD genrealization with datasets sampled from SCM following a different distribution (LIN **OUT**, RFF **OUT**) than the datasets used for training Cond-FiP (LIN **IN**, RFF **IN**). We now analyze how sensitive is Cond-FiP to distribution shifts by comparing its performance across scenarios as the severity of the distribution shift is increased.

To illustrate how we control the magnitude of distribution shift, we discuss the difference in the distribution of causal mechanisms across $\mathbb{P}_{\text{IN}}$ and $\mathbb{P}_{\text{OUT}}$. The distribution shift arises because the support of the parameters of causal mechanisms changes from $\mathbb{P}_{\text{IN}}$ to $\mathbb{P}_{\text{OUT}}$. For example, for linear causal mechanism case, the weights in $\mathbb{P}_{\text{IN}}$ are sampled uniformly from $(-3, -1) \cup (1, 3)$; while in $\mathbb{P}_{\text{OUT}}$ they are sampled from uniformly from $(0.5, 4)$. We now change the support set of the parameters in $\mathbb{P}_{\text{OUT}}$ to $(0.5\alpha, 4\alpha)$, so that by increasing $\alpha$ we make the distribution shift more severe. We follow this procedure for the support set of all the parameters associated with functional mechanisms and generate distributions ($\mathbb{P}_{\text{OUT}}(\alpha)$) with varying shift w.r.t $\mathbb{P}_{\text{IN}}$ by changing $\alpha$. Note that $\alpha = 1$ corresponds to the same $\mathbb{P}_{\text{OUT}}$ as the one used for sampling datasets in our main results.

We conduct two experiments for evaluating the robustness of Cond-FiP to distribution shifts, described ahead.

- **Controlling Shift in Causal Mechanisms.** We start with the parameter configuration of $\mathbb{P}_{\text{OUT}}$ from the setup in main results; and then control the magnitude of shift by changing the support set of parameters of causal mechanisms.

- **Controlling Shift in Noise Variables.** We start with the parameter configuration of $\mathbb{P}_{\text{OUT}}$ from the setup in main results; and then control the magnitude of shift by changing the support set of parameters of noise distribution.

Tables 4, 5, and 6 provide results for the case of controlling shift via causal mechanisms, for the task of noise prediction, sample generation, and interventional generation respectively. We find that the performance of Cond-FiP does not change much as we increase $\alpha$, indicating that Cond-FiP is robust to the varying levels of distribution shits in causal mechanisms.

However, for the case of controlling shift via noise variables (Table 7, 8, and 9) we find that Cond-FiP is quite sensitive to the varying levels of distribution shift in noise variables. The performance of Cond-FiP degrades with increasing magnitude of the shift ($\alpha$) for all the tasks.

| Total Nodes | Shift Level ($\alpha$) | LIN **OUT** | RFF **OUT** |
|:---:|:---:|:---:|:---:|
| 10 | 1 | 0.07 (0.01) | 0.10 (0.01) |
| 10 | 2 | 0.06 (0.01) | 0.10 (0.01) |
| 10 | 5 | 0.05 (0.01) | 0.10 (0.01) |
| 10 | 10 | 0.05 (0.01) | 0.10 (0.01) |
| 20 | 1 | 0.07 (0.0) | 0.12 (0.0) |
| 20 | 2 | 0.06 (0.0) | 0.13 (0.01) |
| 20 | 5 | 0.05 (0.0) | 0.11 (0.01) |
| 20 | 10 | 0.05 (0.0) | 0.10 (0.01) |
| 50 | 1 | 0.07 (0.01) | 0.14 (0.01) |
| 50 | 2 | 0.05 (0.01) | 0.17 (0.01) |
| 50 | 5 | 0.05 (0.01) | 0.14 (0.01) |
| 50 | 10 | 0.04 (0.0) | 0.14 (0.01) |
| 100 | 1 | 0.07 (0.01) | 0.16 (0.01) |
| 100 | 2 | 0.05 (0.01) | 0.18 (0.0) |
| 100 | 5 | 0.05 (0.0) | 0.17 (0.01) |
| 100 | 10 | 0.05 (0.0) | 0.16 (0.01) |

Table 4: **Results for Noise Prediction under Distribution Shifts in Causal Mechanisms.** We evaluate the robustness of Cond-FiP to distribution shifts in the parametrization of causal mechanisms. We vary the distribution shift controlled by $\alpha$, where $\alpha = 1$ corresponds to the results in Table 1. Each cell reports the mean (standard error) RMSE over the multiple test datasets for each scenario. *We find that Cond-FiP is robust to varying levels of distribution shift in causal mechanisms.*

| Total Nodes | Shift Level ($\alpha$) | LIN **OUT** | RFF **OUT** |
|:---:|:---:|:---:|:---:|
| 10 | 1 | 0.05 (0.01) | 0.08 (0.01) |
| 10 | 2 | 0.05 (0.0) | 0.07 (0.01) |
| 10 | 5 | 0.05 (0.0) | 0.07 (0.01) |
| 10 | 10 | 0.06 (0.0) | 0.06 (0.01) |
| 20 | 1 | 0.07 (0.01) | 0.30 (0.03) |
| 20 | 2 | 0.06 (0.01) | 0.34 (0.05) |
| 20 | 5 | 0.06 (0.01) | 0.35 (0.05) |
| 20 | 10 | 0.06 (0.01) | 0.29 (0.07) |
| 50 | 1 | 0.07 (0.0) | 0.48 (0.07) |
| 50 | 2 | 0.07 (0.0) | 0.47 (0.07) |
| 50 | 5 | 0.07 (0.01) | 0.38 (0.06) |
| 50 | 10 | 0.07 (0.01) | 0.32 (0.06) |
| 100 | 1 | 0.09 (0.01) | 0.57 (0.07) |
| 100 | 2 | 0.09 (0.01) | 0.60 (0.05) |
| 100 | 5 | 0.09 (0.01) | 0.58 (0.05) |
| 100 | 10 | 0.12 (0.02) | 0.56 (0.06) |

Table 5: **Results for Sample Generation under Distribution Shifts in Causal Mechanisms.** We evaluate the robustness of Cond-FiP to distribution shifts in the parametrization of causal mechanisms. We vary the distribution shift controlled by $\alpha$, where $\alpha = 1$ corresponds to the results in Table 2. Each cell reports the mean (standard error) RMSE over the multiple test datasets for each scenario. *We find that Cond-FiP is robust to varying levels of distribution shift in causal mechanisms.*

| Total Nodes | Shift Level ($\alpha$) | LIN **OUT** | RFF **OUT** |
|---|---|---|---|
| 10 | 1 | 0.07 (0.01) | 0.11 (0.01) |
| 10 | 2 | 0.07 (0.01) | 0.11 (0.01) |
| 10 | 5 | 0.07 (0.01) | 0.10 (0.01) |
| 10 | 10 | 0.06 (0.01) | 0.10 (0.01) |
| 20 | 1 | 0.14 (0.03) | 0.31 (0.03) |
| 20 | 2 | 0.10 (0.02) | 0.33 (0.04) |
| 20 | 5 | 0.17 (0.1) | 0.34 (0.04) |
| 20 | 10 | 0.10 (0.03) | 0.28 (0.05) |
| 50 | 1 | 0.12 (0.02) | 0.48 (0.07) |
| 50 | 2 | 0.12 (0.03) | 0.47 (0.07) |
| 50 | 5 | 0.11 (0.01) | 0.39 (0.06) |
| 50 | 10 | 0.11 (0.02) | 0.32 (0.06) |
| 100 | 1 | 0.14 (0.02) | 0.58 (0.07) |
| 100 | 2 | 0.13 (0.02) | 0.60 (0.06) |
| 100 | 5 | 0.14 (0.03) | 0.58 (0.05) |
| 100 | 10 | 0.18 (0.04) | 0.55 (0.06) |

Table 6: **Results for Interventional Generation under Distribution Shifts in Causal Mechanisms.** We evaluate the robustness of Cond-FiP to distribution shifts in the parametrization of causal mechanisms. We vary the distribution shift controlled by $\alpha$, where $\alpha = 1$ corresponds to the results in Table 3. Each cell reports the mean (standard error) RMSE over the multiple test datasets for each scenario. *We find that Cond-FiP is robust to varying levels of distribution shift in causal mechanisms.*

| Total Nodes | Shift Level ($\alpha$) | LIN **OUT** | RFF **OUT** |
|---|---|---|---|
| 10 | 1 | 0.07 (0.01) | 0.10 (0.01) |
| 10 | 2 | 0.07 (0.01) | 0.11 (0.01) |
| 10 | 5 | 0.07 (0.01) | 0.18 (0.02) |
| 10 | 10 | 0.08 (0.01) | 0.26 (0.04) |
| 20 | 1 | 0.07 (0.0) | 0.12 (0.0) |
| 20 | 2 | 0.07 (0.0) | 0.16 (0.01) |
| 20 | 5 | 0.07 (0.0) | 0.30 (0.01) |
| 20 | 10 | 0.07 (0.0) | 0.41 (0.02) |
| 50 | 1 | 0.07 (0.01) | 0.14 (0.01) |
| 50 | 2 | 0.07 (0.01) | 0.19 (0.01) |
| 50 | 5 | 0.07 (0.01) | 0.33 (0.02) |
| 50 | 10 | 0.07 (0.01) | 0.44 (0.02) |
| 100 | 1 | 0.07 (0.01) | 0.16 (0.01) |
| 100 | 2 | 0.07 (0.01) | 0.22 (0.0) |
| 100 | 5 | 0.07 (0.01) | 0.35 (0.01) |
| 100 | 10 | 0.07 (0.01) | 0.44 (0.01) |

Table 7: **Results for Noise Prediction under Distribution Shifts in Noise Variables.** We evaluate the robustness of Cond-FiP to distribution shifts in the parametrization of noise distribution. We vary the distribution shift controlled by $\alpha$, where $\alpha = 1$ corresponds to the results in Table 1. Each cell reports the mean (standard error) RMSE over the multiple test datasets for each scenario. *We find that Cond-FiP is sensitive to varying levels of distribution shift in noise variables, its performance decreases with increasing magnitude of the shift.*

| Total Nodes | Shift Level ($\alpha$) | LIN **OUT** | RFF **OUT** |
|:---:|:---:|:---:|:---:|
| 10 | 1 | 0.05 (0.01) | 0.08 (0.01) |
| 10 | 2 | 0.05 (0.0) | 0.13 (0.03) |
| 10 | 5 | 0.05 (0.01) | 0.28 (0.06) |
| 10 | 10 | 0.05 (0.01) | 0.36 (0.08) |
| 20 | 1 | 0.07 (0.01) | 0.30 (0.03) |
| 20 | 2 | 0.07 (0.01) | 0.45 (0.04) |
| 20 | 5 | 0.07 (0.01) | 0.59 (0.03) |
| 20 | 10 | 0.07 (0.01) | 0.58 (0.02) |
| 50 | 1 | 0.07 (0.0) | 0.48 (0.07) |
| 50 | 2 | 0.07 (0.0) | 0.59 (0.06) |
| 50 | 5 | 0.07 (0.0) | 0.64 (0.03) |
| 50 | 10 | 0.07 (0.0) | 0.58 (0.02) |
| 100 | 1 | 0.09 (0.01) | 0.57 (0.07) |
| 100 | 2 | 0.09 (0.01) | 0.63 (0.05) |
| 100 | 5 | 0.09 (0.01) | 0.65 (0.03) |
| 100 | 10 | 0.09 (0.01) | 0.59 (0.02) |

Table 8: **Results for Sample Generation under Distribution Shifts in Noise Variables.** We evaluate the robustness of Cond-FiP to distribution shifts in the parametrization of noise distribution. We vary the distribution shift controlled by $\alpha$, where $\alpha = 1$ corresponds to the results in Table 2. Each cell reports the mean (standard error) RMSE over the multiple test datasets for each scenario. *We find that Cond-FiP is sensitive to varying levels of distribution shift in noise variables, its performance decreases with increasing magnitude of the shift.*

| Total Nodes | Shift Level ($\alpha$) | LIN **OUT** | RFF **OUT** |
|:---:|:---:|:---:|:---:|
| 10 | 1 | 0.07 (0.01) | 0.11 (0.01) |
| 10 | 2 | 0.07 (0.01) | 0.14 (0.02) |
| 10 | 5 | 0.07 (0.01) | 0.25 (0.05) |
| 10 | 10 | 0.07 (0.01) | 0.32 (0.06) |
| 20 | 1 | 0.14 (0.03) | 0.31 (0.03) |
| 20 | 2 | 0.14 (0.03) | 0.42 (0.03) |
| 20 | 5 | 0.14 (0.03) | 0.57 (0.03) |
| 20 | 10 | 0.14 (0.03) | 0.56 (0.02) |
| 50 | 1 | 0.12 (0.02) | 0.48 (0.07) |
| 50 | 2 | 0.12 (0.01) | 0.58 (0.06) |
| 50 | 5 | 0.12 (0.01) | 0.65 (0.04) |
| 50 | 10 | 0.12 (0.01) | 0.59 (0.02) |
| 100 | 1 | 0.14 (0.02) | 0.58 (0.07) |
| 100 | 2 | 0.14 (0.02) | 0.65 (0.06) |
| 100 | 5 | 0.14 (0.02) | 0.67 (0.04) |
| 100 | 10 | 0.14 (0.02) | 0.60 (0.03) |

Table 9: **Results for Interventional Generation under Distribution Shifts in Noise Variables.** We evaluate the robustness of Cond-FiP to distribution shifts in the parametrization of noise distribution. We vary the distribution shift controlled by $\alpha$, where $\alpha = 1$ corresponds to the results in Table 3. Each cell reports the mean (standard error) RMSE over the multiple test datasets for each scenario. *We find that Cond-FiP is sensitive to varying levels of distribution shift in noise variables, its performance decreases with increasing magnitude of the shift.*

# E    Experiment on Generalization in Scarce Data Regime on AVICI benchmark

## E.1    Experiments with $n_{\mathcal{D}_{\text{test}}} = 100$

In this section we benchmark Cond-FiP against the baselines for the scenario when test datasets in the input context have smaller sample size ($n_{\mathcal{D}_{\text{test}}} = 100$) as compared to the train datasets ($n_{\mathcal{D}_{\text{test}}} = 400$) in Appendix C.

We report the results for the task of noise prediction, sample generation, and interventional generation in Table 10, Table 11, and Table 12 respectively. We find that Cond-FiP exhibits superior generalization as compared to baselines. For example, in the case of RFF **IN**, Cond-FiP is even better than FiP for all the tasks! This can be attributed to the advantage of amortized inference; as the sample size in test dataset decreases, the generalization of baselines would be affected a lot since they require training from scratch on these datasets. However, amortized inference methods would be impacted less as they do not have to trained from scratch, and the inductive bias learned by them can help them generalize even with smaller input context.

| Method | Total Nodes | LIN **IN** | RFF **IN** | LIN **OUT** | RFF **OUT** |
|---|---|---|---|---|---|
| DoWhy | 10 | 0.06 (0.01) | 0.22 (0.03) | 0.09 (0.01) | 0.16 (0.03) |
| DECI | 10 | 0.15 (0.01) | 0.3 (0.02) | 0.22 (0.01) | 0.3 (0.03) |
| FiP | 10 | 0.07 (0.01) | 0.18 (0.01) | 0.12 (0.01) | 0.11 (0.01) |
| Cond-FiP | 10 | 0.07 (0.01) | 0.14 (0.01) | 0.09 (0.01) | 0.14 (0.01) |
| DoWhy | 20 | 0.06 (0.01) | 0.27 (0.05) | 0.07 (0.01) | 0.37 (0.01) |
| DECI | 20 | 0.15 (0.02) | 0.33 (0.02) | 0.17 (0.02) | 0.35 (0.03) |
| FiP | 20 | 0.09 (0.01) | 0.21 (0.03) | 0.1 (0.01) | 0.27 (0.03) |
| Cond-FiP | 20 | 0.08 (0.01) | 0.12 (0.01) | 0.1 (0.01) | 0.15 (0.01) |
| DoWhy | 50 | 0.06 (0.01) | 0.29 (0.04) | 0.05 (0.01) | 0.47 (0.04) |
| DECI | 50 | 0.14 (0.01) | 0.33 (0.02) | 0.14 (0.02) | 0.4 (0.03) |
| FiP | 50 | 0.08 (0.01) | 0.23 (0.03) | 0.08 (0.01) | 0.37 (0.04) |
| Cond-FiP | 50 | 0.08 (0.0) | 0.12 (0.01) | 0.08 (0.01) | 0.15 (0.01) |
| DoWhy | 100 | 0.06 (0.01) | 0.31 (0.04) | 0.06 (0.01) | 0.5 (0.03) |
| DECI | 100 | 0.13 (0.01) | 0.36 (0.03) | 0.12 (0.02) | 0.44 (0.02) |
| FiP | 100 | 0.08 (0.01) | 0.25 (0.04) | 0.1 (0.01) | 0.39 (0.03) |
| Cond-FiP | 100 | 0.07 (0.0) | 0.13 (0.01) | 0.08 (0.01) | 0.17 (0.01) |

Table 10: **Results for Noise Prediction with Smaller Sample Size ($n_{\mathcal{D}_{\text{test}}} = 100$).** We compare Cond-FiP against the baselines for the task of predicting noise variable from input observations. Each test dataset contains 100 samples, as opposed to 400 samples in Table 1. Each cell reports the mean (standard error) RMSE over the multiple test datasets for each scenario. Shaded rows deonte the case where the graph size is larger than the train graph sizes ($d = 20$) for Cond-FiP. *Results show that Cond-FiP generalizes much better than the baselines in this low-data regime.*

| Method | Total Nodes | LIN **IN** | RFF **IN** | LIN **OUT** | RFF **OUT** |
|---|---|---|---|---|---|
| DoWhy | 10 | 0.1 (0.01) | 0.3 (0.06) | 0.12 (0.02) | 0.19 (0.03) |
| DECI | 10 | 0.23 (0.01) | 0.45 (0.04) | 0.31 (0.02) | 0.38 (0.04) |
| FiP | 10 | 0.13 (0.01) | 0.29 (0.04) | 0.18 (0.02) | 0.15 (0.03) |
| Cond-FiP | 10 | 0.09 (0.01) | 0.2 (0.03) | 0.09 (0.02) | 0.14 (0.02) |
| DoWhy | 20 | 0.11 (0.01) | 0.47 (0.15) | 0.11 (0.02) | 0.5 (0.03) |
| DECI | 20 | 0.26 (0.02) | 0.53 (0.05) | 0.26 (0.03) | 0.57 (0.04) |
| FiP | 20 | 0.17 (0.02) | 0.34 (0.06) | 0.17 (0.02) | 0.39 (0.03) |
| Cond-FiP | 20 | 0.08 (0.0) | 0.31 (0.06) | 0.13 (0.01) | 0.37 (0.02) |
| DoWhy | 50 | 0.11 (0.01) | 0.42 (0.08) | 0.09 (0.01) | 0.66 (0.06) |
| DECI | 50 | 0.23 (0.02) | 0.59 (0.08) | 0.27 (0.04) | 0.73 (0.06) |
| FiP | 50 | 0.13 (0.01) | 0.38 (0.07) | 0.14 (0.01) | 0.58 (0.06) |
| Cond-FiP | 50 | 0.1 (0.01) | 0.32 (0.05) | 0.12 (0.01) | 0.54 (0.05) |
| DoWhy | 100 | 0.11 (0.01) | 0.44 (0.08) | 0.11 (0.01) | 0.74 (0.05) |
| DECI | 100 | 0.25 (0.02) | 0.62 (0.08) | 0.25 (0.01) | 0.78 (0.07) |
| FiP | 100 | 0.15 (0.01) | 0.4 (0.07) | 0.19 (0.02) | 0.67 (0.07) |
| Cond-FiP | 100 | 0.11 (0.01) | 0.35 (0.07) | 0.14 (0.02) | 0.63 (0.07) |

Table 11: **Results for Sample Generation with Smaller Sample Size ($n_{\mathcal{D}_{\text{test}}} = 100$).** We compare Cond-FiP against the baselines for the task of generating samples from the input noise variable. Each test dataset contains 100 samples, as opposed to 400 samples in Table 2. Each cell reports the mean (standard error) RMSE over the multiple test datasets for each scenario. Shaded rows deonte the case where the graph size is larger than the train graph sizes ($d = 20$) for Cond-FiP. *Results show that Cond-FiP generalizes much better than the baselines in this low-data regime.*

| Method | Total Nodes | LIN **IN** | RFF **IN** | LIN **OUT** | RFF **OUT** |
|---|---|---|---|---|---|
| DoWhy | 10 | 0.09 (0.01) | 0.34 (0.08) | 0.11 (0.01) | 0.2 (0.04) |
| DECI | 10 | 0.24 (0.02) | 0.43 (0.04) | 0.26 (0.03) | 0.35 (0.04) |
| FiP | 10 | 0.13 (0.01) | 0.29 (0.04) | 0.14 (0.02) | 0.14 (0.03) |
| Cond-FiP | 10 | 0.09 (0.02) | 0.21 (0.03) | 0.09 (0.01) | 0.12 (0.02) |
| DoWhy | 20 | 0.1 (0.01) | 0.37 (0.08) | 0.11 (0.02) | 0.49 (0.04) |
| DECI | 20 | 0.25 (0.03) | 0.5 (0.05) | 0.28 (0.03) | 0.54 (0.04) |
| FiP | 20 | 0.16 (0.01) | 0.33 (0.06) | 0.2 (0.03) | 0.38 (0.03) |
| Cond-FiP | 20 | 0.1 (0.01) | 0.27 (0.05) | 0.15 (0.02) | 0.29 (0.03) |
| DoWhy | 50 | 0.12 (0.02) | 0.49 (0.14) | 0.09 (0.01) | 0.64 (0.07) |
| DECI | 50 | 0.26 (0.03) | 0.56 (0.07) | 0.26 (0.03) | 0.72 (0.06) |
| FiP | 50 | 0.16 (0.02) | 0.36 (0.06) | 0.15 (0.01) | 0.57 (0.06) |
| Cond-FiP | 50 | 0.13 (0.02) | 0.29 (0.04) | 0.12 (0.01) | 0.49 (0.07) |
| DoWhy | 100 | 0.11 (0.01) | 0.46 (0.07) | 0.11 (0.01) | 1.16 (0.38) |
| DECI | 100 | 0.24 (0.02) | 0.62 (0.08) | 0.26 (0.01) | 0.78 (0.07) |
| FiP | 100 | 0.16 (0.02) | 0.39 (0.07) | 0.2 (0.02) | 0.66 (0.07) |
| Cond-FiP | 100 | 0.12 (0.02) | 0.32 (0.07) | 0.13 (0.01) | 0.58 (0.07) |

Table 12: **Results for Interventional Generation with Smaller Sample Size ($n_{\mathcal{D}_{\text{test}}} = 100$).** We compare Cond-FiP against the baselines for the task of generating interventional data from the input noise variable. Each test dataset contains 100 samples, as opposed to 400 samples in Table 3. Each cell reports the mean (standard error) RMSE over the multiple test datasets for each scenario. Shaded rows deonte the case where the graph size is larger than the train graph sizes ($d = 20$) for Cond-FiP. *Results show that Cond-FiP generalizes much better than the baselines in this low-data regime.*

 **E.2   Experiments with** $n_{\mathcal{D}_{\text{test}}} = 50$

1002 We conduct more experiments for the smaller sample size scenarios, where decrease the sample size
1003 even further to $n_{\mathcal{D}_{\text{test}}} = 50$ samples. We report the results for the task of noise prediction, sample
1004 generation, and interventional generation in Table 13, Table 14, and Table 15 respectively. We find
1005 that baselines perform much worse than Cond-FiP for the all different SCM distributions, highlighting
1006 the efficacy of Cond-FiP for inferring causal mechanisms when the input context has smaller sample
1007 size. Note that there were issues with training DoWhy for such a small dataset, hence we do not
1008 consider them for this scenario.

| Method | Total Nodes | LIN **IN** | RFF **IN** | LIN **OUT** | RFF **OUT** |
|---|---|---|---|---|---|
| DECI | 10 | 0.19 (0.02) | 0.41 (0.03) | 0.2 (0.02) | 0.42 (0.04) |
| FiP | 10 | 0.13 (0.03) | 0.27 (0.03) | 0.15 (0.02) | 0.21 (0.03) |
| Cond-FiP | 10 | 0.09 (0.01) | 0.17 (0.01) | 0.11 (0.01) | 0.16 (0.01) |
| DECI | 20 | 0.2 (0.01) | 0.42 (0.03) | 0.25 (0.04) | 0.45 (0.05) |
| FiP | 20 | 0.12 (0.01) | 0.33 (0.04) | 0.15 (0.02) | 0.35 (0.04) |
| Cond-FiP | 20 | 0.1 (0.01) | 0.16 (0.01) | 0.11 (0.01) | 0.17 (0.01) |
| DECI | 50 | 0.2 (0.02) | 0.43 (0.02) | 0.2 (0.03) | 0.5 (0.05) |
| FiP | 50 | 0.13 (0.01) | 0.32 (0.03) | 0.13 (0.01) | 0.49 (0.05) |
| Cond-FiP | 50 | 0.1 (0.01) | 0.16 (0.0) | 0.1 (0.01) | 0.17 (0.01) |
| DECI | 100 | 0.19 (0.02) | 0.43 (0.03) | 0.21 (0.01) | 0.53 (0.02) |
| FiP | 100 | 0.11 (0.01) | 0.32 (0.04) | 0.13 (0.01) | 0.48 (0.02) |
| Cond-FiP | 100 | 0.09 (0.01) | 0.16 (0.01) | 0.09 (0.01) | 0.18 (0.01) |

Table 13: **Results for Noise Prediction with Smaller Sample Size (**$n_{\mathcal{D}_{\text{test}}} = 50$**).** We compare
Cond-FiP against the baselines for the task of predicting noise variable from input observations. Each
test dataset contains 50 samples, as opposed to 400 samples in Table 1. Each cell reports the mean
(standard error) RMSE over the multiple test datasets for each scenario. Shaded rows denote the case
where the graph size is larger than the train graph sizes ($d = 20$) for Cond-FiP. *Results show that
Cond-FiP generalizes much better than the baselines in this low-data regime.*

| Method | Total Nodes | LIN **IN** | RFF **IN** | LIN **OUT** | RFF **OUT** |
|---|---|---|---|---|---|
| DECI | 10 | 0.31 (0.02) | 0.58 (0.05) | 0.27 (0.04) | 0.49 (0.07) |
| FiP | 10 | 0.2 (0.03) | 0.4 (0.05) | 0.21 (0.03) | 0.25 (0.04) |
| Cond-FiP | 10 | 0.12 (0.02) | 0.28 (0.03) | 0.12 (0.01) | 0.18 (0.03) |
| DECI | 20 | 0.34 (0.02) | 0.66 (0.08) | 0.39 (0.07) | 0.68 (0.05) |
| FiP | 20 | 0.2 (0.01) | 0.51 (0.08) | 0.25 (0.04) | 0.51 (0.02) |
| Cond-FiP | 20 | 0.13 (0.01) | 0.4 (0.06) | 0.19 (0.02) | 0.43 (0.02) |
| DECI | 50 | 0.32 (0.02) | 0.66 (0.06) | 0.36 (0.02) | 0.8 (0.06) |
| FiP | 50 | 0.2 (0.01) | 0.48 (0.07) | 0.22 (0.02) | 0.69 (0.06) |
| Cond-FiP | 50 | 0.15 (0.02) | 0.4 (0.05) | 0.16 (0.01) | 0.59 (0.06) |
| DECI | 100 | 0.36 (0.04) | 0.68 (0.08) | 0.39 (0.03) | 0.84 (0.06) |
| FiP | 100 | 0.2 (0.02) | 0.49 (0.09) | 0.28 (0.03) | 0.73 (0.07) |
| Cond-FiP | 100 | 0.16 (0.01) | 0.42 (0.07) | 0.22 (0.01) | 0.65 (0.06) |

Table 14: **Results for Sample Generation with Smaller Sample Size ($n_{\mathcal{D}_{\text{test}}} = 50$).** We compare Cond-FiP against the baselines for the task of generating samples from the input noise variable. Each test dataset contains 50 samples, as opposed to 400 samples in Table 2. Each cell reports the mean (standard error) RMSE over the multiple test datasets for each scenario. Shaded rows denote the case where the graph size is larger than the train graph sizes ($d = 20$) for Cond-FiP. *Results show that Cond-FiP generalizes much better than the baselines in this low-data regime.*

| Method | Total Nodes | LIN **IN** | RFF **IN** | LIN **OUT** | RFF **OUT** |
|---|---|---|---|---|---|
| DECI | 10 | 0.3 (0.03) | 0.53 (0.05) | 0.26 (0.04) | 0.42 (0.05) |
| FiP | 10 | 0.21 (0.04) | 0.35 (0.04) | 0.2 (0.03) | 0.22 (0.03) |
| Cond-FiP | 10 | 0.12 (0.01) | 0.19 (0.03) | 0.07 (0.01) | 0.14 (0.02) |
| DECI | 20 | 0.33 (0.02) | 0.6 (0.06) | 0.43 (0.07) | 0.63 (0.04) |
| FiP | 20 | 0.21 (0.02) | 0.46 (0.07) | 0.29 (0.04) | 0.49 (0.02) |
| Cond-FiP | 20 | 0.11 (0.01) | 0.29 (0.06) | 0.15 (0.02) | 0.32 (0.03) |
| DECI | 50 | 0.34 (0.02) | 0.66 (0.07) | 0.34 (0.02) | 0.78 (0.06) |
| FiP | 50 | 0.21 (0.02) | 0.46 (0.07) | 0.23 (0.02) | 0.68 (0.06) |
| Cond-FiP | 50 | 0.13 (0.02) | 0.31 (0.05) | 0.12 (0.02) | 0.51 (0.07) |
| DECI | 100 | 0.37 (0.04) | 0.67 (0.08) | 0.4 (0.04) | 0.84 (0.06) |
| FiP | 100 | 0.21 (0.02) | 0.49 (0.08) | 0.28 (0.03) | 0.73 (0.07) |
| Cond-FiP | 100 | 0.12 (0.01) | 0.33 (0.07) | 0.14 (0.01) | 0.58 (0.07) |

Table 15: **Results for Interventional Generation with Smaller Sample Size ($n_{\mathcal{D}_{\text{test}}} = 50$).** We compare Cond-FiP against the baselines for the task of generating interventional data from the input noise variable. Each test dataset contains 50 samples, as opposed to 400 samples in Table 3. Each cell reports the mean (standard error) RMSE over the multiple test datasets for each scenario. Shaded rows deonte the case where the graph size is larger than the train graph sizes ($d = 20$) for Cond-FiP. *Results show that Cond-FiP generalizes much better than the baselines in this low-data regime.*

# F    Experiments without True Causal Graph on AVICI Benchmark

Results in Appendix C (Table 1, Table 2, Table 3) require the knowledge of true graph ($\mathcal{G}$) as part of the input context to Cond-FiP. In this section we conduct where we don't provide the true graph in the input context, rather we infer the graph $\hat{\mathcal{G}}$ using an amortized causal discovery approach (AVICI [Lorch et al., 2022]) from the observational data $D_{\boldsymbol{X}}$. We chose AVICI for this task since it can enable to amortized inference of causal graphs, hence allowing the combined pipeline of AVICI + Cond-FiP can perform amortized inference of SCMs. More precisely, AVICI infers the graph from a novel instance $\mathcal{G}$ from input context $D_{\boldsymbol{X}}$ without updating any parameters, and we pass $(\hat{\mathcal{G}}, D_{\boldsymbol{X}})$ as the input context for Cond-FiP. Therefore, for any $\boldsymbol{z} \in \mathbb{R}^d$, Cond-FiP ($\mathcal{T}(\boldsymbol{z}, D_{\boldsymbol{X}}, \hat{\mathcal{G}})$) aims to replicate the functional mechanism $\boldsymbol{F}(\boldsymbol{z})$ of the underlying SCM.

The results for benchmarking Cond-FiP with inferred graphs using AVICI for the task of noise prediction, sample generation, and interventional generation are provided in Table 16, Table 17, and Table 18 respectively. For a fair comparison, the baselines FiP, DECI, and DoWhy also use the inferred graph ($\hat{\mathcal{G}}$) by AVICI instead of the true graph ($\mathcal{G}$). We find that Cond-FiP remains competitive to baselines even for the scenario of unknown true causal graph. Hence, our training procedure can be extended for amortized inference of both causal graphs and causal mechanisms of the SCM.

| Method | Total Nodes | LIN **IN** | RFF **IN** | LIN **OUT** | RFF **OUT** |
| --- | --- | --- | --- | --- | --- |
| DoWhy | 10 | 0.16 (0.05) | 0.24 (0.04) | 0.12 (0.03) | 0.12 (0.02) |
| DECI | 10 | 0.21 (0.05) | 0.29 (0.04) | 0.16 (0.03) | 0.19 (0.04) |
| FiP | 10 | 0.16 (0.05) | 0.2 (0.04) | 0.13 (0.03) | 0.09 (0.01) |
| Cond-FiP | 10 | 0.15 (0.05) | 0.2 (0.04) | 0.13 (0.03) | 0.11 (0.01) |
| DoWhy | 20 | 0.19 (0.05) | 0.22 (0.03) | 0.2 (0.03) | 0.26 (0.01) |
| DECI | 20 | 0.23 (0.05) | 0.28 (0.03) | 0.24 (0.04) | 0.28 (0.02) |
| FiP | 20 | 0.2 (0.05) | 0.2 (0.03) | 0.21 (0.03) | 0.21 (0.02) |
| Cond-FiP | 20 | 0.18 (0.05) | 0.17 (0.02) | 0.21 (0.03) | 0.16 (0.02) |
| DoWhy | 50 | 0.44 (0.05) | 0.3 (0.03) | 0.51 (0.03) | 0.38 (0.04) |
| DECI | 50 | 0.46 (0.05) | 0.33 (0.04) | 0.52 (0.03) | 0.42 (0.05) |
| FiP | 50 | 0.44 (0.05) | 0.28 (0.04) | 0.51 (0.03) | 0.35 (0.05) |
| Cond-FiP | 50 | 0.43 (0.05) | 0.24 (0.03) | 0.53 (0.03) | 0.29 (0.04) |
| DoWhy | 100 | 0.49 (0.06) | 0.38 (0.03) | 0.64 (0.03) | 0.53 (0.04) |
| DECI | 100 | 0.5 (0.06) | 0.41 (0.03) | 0.64 (0.03) | 0.55 (0.03) |
| FiP | 100 | 0.49 (0.06) | 0.37 (0.03) | 0.64 (0.03) | 0.51 (0.04) |
| Cond-FiP | 100 | 0.48 (0.06) | 0.34 (0.03) | 0.64 (0.03) | 0.49 (0.04) |

Table 16: **Results for Noise Prediction without True Graph.** We compare Cond-FiP against the baselines for the task of predicting noise variable from input observations. Unlike experiments in Table 1, the true graph $\mathcal{G}$ is not present in input context, rather its inferred via AVICI [Lorch et al., 2022]. Each cell reports the mean (standard error) RMSE over the multiple test datasets for each scenario. Shaded rows deonte the case where the graph size is larger than the train graph sizes ($d = 20$) for Cond-FiP. *Results indicate Cond-FiP can generalize to novel instances even in the absence of true graph.*

| Method | Total Nodes | LIN **IN** | RFF **IN** | LIN **OUT** | RFF **OUT** |
|---|---|---|---|---|---|
| DoWhy | 10 | 0.22 (0.07) | 0.29 (0.05) | 0.13 (0.04) | 0.14 (0.02) |
| DECI | 10 | 0.29 (0.06) | 0.39 (0.05) | 0.18 (0.04) | 0.22 (0.05) |
| FiP | 10 | 0.23 (0.06) | 0.26 (0.05) | 0.15 (0.04) | 0.12 (0.02) |
| Cond-FiP | 10 | 0.22 (0.07) | 0.26 (0.05) | 0.13 (0.04) | 0.11 (0.02) |
| DoWhy | 20 | 0.25 (0.05) | 0.38 (0.06) | 0.29 (0.06) | 0.42 (0.03) |
| DECI | 20 | 0.3 (0.06) | 0.52 (0.07) | 0.34 (0.06) | 0.47 (0.04) |
| FiP | 20 | 0.26 (0.05) | 0.37 (0.07) | 0.3 (0.06) | 0.33 (0.04) |
| Cond-FiP | 20 | 0.24 (0.05) | 0.36 (0.06) | 0.29 (0.06) | 0.35 (0.03) |
| DoWhy | 50 | 0.53 (0.07) | 0.46 (0.06) | 0.58 (0.03) | 0.59 (0.07) |
| DECI | 50 | 0.55 (0.07) | 0.54 (0.07) | 0.59 (0.02) | 0.66 (0.06) |
| FiP | 50 | 0.53 (0.07) | 0.44 (0.05) | 0.58 (0.02) | 0.53 (0.07) |
| Cond-FiP | 50 | 0.52 (0.07) | 0.43 (0.05) | 0.58 (0.02) | 0.53 (0.07) |
| DoWhy | 100 | 0.67 (0.07) | 0.52 (0.06) | 0.69 (0.02) | 0.68 (0.04) |
| DECI | 100 | 0.69 (0.08) | 0.57 (0.08) | 0.69 (0.02) | 0.71 (0.04) |
| FiP | 100 | 0.66 (0.07) | 0.5 (0.07) | 0.68 (0.02) | 0.64 (0.05) |
| Cond-FiP | 100 | 0.64 (0.06) | 0.49 (0.06) | 0.68 (0.02) | 0.63 (0.05) |

Table 17: **Results for Sample Generation without True Graph.** We compare Cond-FiP against the baselines for the task of generating samples from the input noise variable. Unlike experiments in Table 2, the true graph $\mathcal{G}$ is not present in input context, rather its inferred via AVICI [Lorch et al., 2022].. Each cell reports the mean (standard error) RMSE over the multiple test datasets for each scenario. Shaded rows deonte the case where the graph size is larger than the train graph sizes ($d = 20$) for Cond-FiP. *Results indicate Cond-FiP can generalize to novel instances even in the absence of true graph.*

| Method | Total Nodes | LIN **IN** | RFF **IN** | LIN **OUT** | RFF **OUT** |
|---|---|---|---|---|---|
| DoWhy | 10 | 0.32 (0.09) | 0.3 (0.05) | 0.13 (0.04) | 0.13 (0.02) |
| DECI | 10 | 0.37 (0.08) | 0.39 (0.05) | 0.17 (0.03) | 0.21 (0.04) |
| FiP | 10 | 0.32 (0.08) | 0.27 (0.05) | 0.14 (0.04) | 0.1 (0.02) |
| Cond-FiP | 10 | 0.31 (0.08) | 0.3 (0.05) | 0.14 (0.04) | 0.13 (0.02) |
| DoWhy | 20 | 0.29 (0.06) | 0.38 (0.07) | 0.37 (0.05) | 0.4 (0.03) |
| DECI | 20 | 0.34 (0.06) | 0.51 (0.07) | 0.41 (0.05) | 0.43 (0.03) |
| FiP | 20 | 0.3 (0.06) | 0.37 (0.07) | 0.38 (0.05) | 0.31 (0.03) |
| Cond-FiP | 20 | 0.29 (0.06) | 0.37 (0.06) | 0.37 (0.05) | 0.33 (0.03) |
| DoWhy | 50 | 0.54 (0.08) | 0.45 (0.06) | 0.62 (0.04) | 0.57 (0.06) |
| DECI | 50 | 0.57 (0.08) | 0.52 (0.07) | 0.63 (0.03) | 0.64 (0.06) |
| FiP | 50 | 0.55 (0.08) | 0.43 (0.05) | 0.62 (0.03) | 0.51 (0.07) |
| Cond-FiP | 50 | 0.54 (0.08) | 0.43 (0.05) | 0.62 (0.03) | 0.51 (0.06) |
| DoWhy | 100 | 0.66 (0.06) | 0.52 (0.07) | 0.71 (0.05) | 0.65 (0.05) |
| DECI | 100 | 0.68 (0.07) | 0.58 (0.09) | 0.71 (0.05) | 0.7 (0.04) |
| FiP | 100 | 0.65 (0.06) | 0.51 (0.07) | 0.71 (0.05) | 0.62 (0.05) |
| Cond-FiP | 100 | 0.64 (0.06) | 0.49 (0.06) | 0.7 (0.04) | 0.62 (0.05) |

Table 18: **Results for Interventional Generation without True Graph.** We compare Cond-FiP against the baselines for the task of interventional data from the input noise variable. Unlike experiments in Table 3, the true graph $\mathcal{G}$ is not present in input context, rather its inferred via AVICI [Lorch et al., 2022]. Each cell reports the mean (standard error) RMSE over the multiple test datasets for each scenario. Shaded rows deonte the case where the graph size is larger than the train graph sizes ($d = 20$) for Cond-FiP. *Results indicate Cond-FiP can generalize to novel instances even in the absence of true graph.*

 # G    Ablation Study on AVICI benchmark

 ## G.1    Ablation Study of Encoder

We conduct an ablation study where we train two variants of the encoder in Cond-FiP described as follows:

- *Cond-FiP (LIN)*: We sample SCMs with linear causal mechanisms during training of the encoder.
- *Cond-FiP (RFF)*: We sample SCMs with non-linear causal mechanisms during training of the encoder.

Note that for the training the subsequent decoder, we sample SCMs with both linear and rff causal mechanisms as in the main results ( Table 1, Table 2, and Table 3). Note that in the main results, the encoder was trained by sampling SCMs with both linear and rff functional relationships. Hence, this ablation helps us to understand whether the strategy of training encoder on mixed functional relationships can bring more generalization to the amortization process, or if we should have trained encoders specialized for linear and non-linear functional relationships.

We present our results of the ablation study for the task of noise prediction, sample generation, and interventional generation in Table 19, Table 20, Table 21 respectively. Our findings indicate that Cond-FiP is robust to the choice of encoder training strategy! Even though the encoder for Cond-FiP (RFF) was only trained on data from non-linear SCMs, its generalization performance is similar to Cond-FiP where the encoder was trained on data from both linear and non-linear SCMs.

| Method | Total Nodes | LIN **IN** | RFF **IN** | LIN **OUT** | RFF **OUT** |
|---|---|---|---|---|---|
| Cond-FiP(LIN) | 10 | 0.07 (0.01) | 0.21 (0.02) | 0.08 (0.01) | 0.2 (0.03) |
| Cond-FiP(RFF) | 10 | 0.06 (0.01) | 0.11 (0.01) | 0.07 (0.01) | 0.09 (0.01) |
| Cond-FiP | 10 | 0.06 (0.01) | 0.1 (0.01) | 0.07 (0.01) | 0.1 (0.01) |
| Cond-FiP(LIN) | 20 | 0.07 (0.01) | 0.19 (0.02) | 0.09 (0.01) | 0.21 (0.01) |
| Cond-FiP(RFF) | 20 | 0.06 (0.01) | 0.09 (0.01) | 0.1 (0.02) | 0.11 (0.01) |
| Cond-FiP | 20 | 0.06 (0.01) | 0.09 (0.01) | 0.07 (0.0) | 0.12 (0.0) |
| Cond-FiP(LIN) | 50 | 0.07 (0.01) | 0.21 (0.02) | 0.07 (0.01) | 0.24 (0.01) |
| Cond-FiP(RFF) | 50 | 0.07 (0.01) | 0.09 (0.01) | 0.07 (0.0) | 0.14 (0.01) |
| Cond-FiP | 50 | 0.06 (0.01) | 0.1 (0.01) | 0.07 (0.01) | 0.14 (0.01) |
| Cond-FiP(LIN) | 100 | 0.06 (0.0) | 0.22 (0.02) | 0.07 (0.01) | 0.26 (0.01) |
| Cond-FiP(RFF) | 100 | 0.06 (0.01) | 0.09 (0.01) | 0.07 (0.01) | 0.14 (0.01) |
| Cond-FiP | 100 | 0.05 (0.0) | 0.1 (0.01) | 0.07 (0.01) | 0.16 (0.01) |

Table 19: **Encoder Ablation for Noise Prediction.** We compare Cond-FiP against the baselines for the task of predicting noise variable from input observations against two variants. One variant corresponds to the encoder trained on SCMs with only linear functional relationships, Cond-FiP(LIN). Similarly, we have another variant where the decoder was trained on SCMs with only rff functional relationships, Cond-FiP(RFF). Each cell reports the mean (standard error) RMSE over the multiple test datasets for each scenario. *Results show that training on only non-linear SCMs* (Cond-FiP(RFF)) *gives similar performance as training on both linear and non-linear SCMs* (Cond-FiP).

| Method | Total Nodes | LIN **IN** | RFF **IN** | LIN **OUT** | RFF **OUT** |
|---|---|---|---|---|---|
| Cond-FiP(LIN) | 10 | 0.05 (0.01) | 0.14 (0.02) | 0.06 (0.0) | 0.08 (0.01) |
| Cond-FiP(RFF) | 10 | 0.08 (0.01) | 0.18 (0.06) | 0.06 (0.0) | 0.07 (0.01) |
| Cond-FiP | 10 | 0.06 (0.01) | 0.14 (0.02) | 0.05 (0.01) | 0.08 (0.01) |
| Cond-FiP(LIN) | 20 | 0.05 (0.01) | 0.25 (0.06) | 0.07 (0.01) | 0.3 (0.03) |
| Cond-FiP(RFF) | 20 | 0.08 (0.01) | 0.22 (0.05) | 0.11 (0.01) | 0.29 (0.03) |
| Cond-FiP | 20 | 0.05 (0.01) | 0.24 (0.06) | 0.07 (0.01) | 0.3 (0.03) |
| Cond-FiP(LIN) | 50 | 0.08 (0.01) | 0.26 (0.05) | 0.11 (0.04) | 0.52 (0.08) |
| Cond-FiP(RFF) | 50 | 0.11 (0.01) | 0.26 (0.05) | 0.15 (0.02) | 0.48 (0.07) |
| Cond-FiP | 50 | 0.08 (0.01) | 0.25 (0.05) | 0.07 (0.0) | 0.48 (0.07) |
| Cond-FiP(LIN) | 100 | 0.07 (0.01) | 0.27 (0.06) | 0.08 (0.0) | 0.57 (0.07) |
| Cond-FiP(RFF) | 100 | 0.11 (0.01) | 0.29 (0.08) | 0.18 (0.03) | 0.61 (0.08) |
| Cond-FiP | 100 | 0.07 (0.01) | 0.29 (0.07) | 0.09 (0.01) | 0.57 (0.07) |

Table 20: **Encoder Ablation for Sample Generation.** We compare Cond-FiP against the baselines for the task of generating samples from input noise variables against two variants. One variant corresponds to the encoder trained on SCMs with only linear functional relationships, Cond-FiP(LIN). Similarly, we have another variant where the decoder was trained on SCMs with only rff functional relationships, Cond-FiP(RFF). Each cell reports the mean (standard error) RMSE over the multiple test datasets for each scenario. *Results show that training on only non-linear SCMs* (Cond-FiP(RFF)) *gives similar performance as training on both linear and non-linear SCMs* (Cond-FiP).

| Method | Total Nodes | LIN **IN** | RFF **IN** | LIN **OUT** | RFF **OUT** |
|---|---|---|---|---|---|
| Cond-FiP(LIN) | 10 | 0.09 (0.02) | 0.2 (0.03) | 0.06 (0.01) | 0.1 (0.01) |
| Cond-FiP(RFF) | 10 | 0.13 (0.04) | 0.23 (0.08) | 0.08 (0.01) | 0.1 (0.01) |
| Cond-FiP | 10 | 0.1 (0.03) | 0.21 (0.03) | 0.07 (0.01) | 0.11 (0.01) |
| Cond-FiP(LIN) | 20 | 0.08 (0.01) | 0.24 (0.05) | 0.12 (0.04) | 0.3 (0.03) |
| Cond-FiP(RFF) | 20 | 0.13 (0.02) | 0.23 (0.05) | 0.13 (0.03) | 0.31 (0.02) |
| Cond-FiP | 20 | 0.09 (0.01) | 0.24 (0.05) | 0.14 (0.03) | 0.31 (0.03) |
| Cond-FiP(LIN) | 50 | 0.12 (0.02) | 0.29 (0.05) | 0.1 (0.01) | 0.51 (0.07) |
| Cond-FiP(RFF) | 50 | 0.14 (0.02) | 0.29 (0.05) | 0.18 (0.03) | 0.47 (0.06) |
| Cond-FiP | 50 | 0.13 (0.02) | 0.27 (0.04) | 0.12 (0.02) | 0.48 (0.07) |
| Cond-FiP(LIN) | 100 | 0.1 (0.01) | 0.3 (0.06) | 0.12 (0.01) | 0.56 (0.07) |
| Cond-FiP(RFF) | 100 | 0.12 (0.01) | 0.31 (0.07) | 0.2 (0.04) | 0.6 (0.09) |
| Cond-FiP | 100 | 0.1 (0.01) | 0.3 (0.06) | 0.14 (0.02) | 0.58 (0.07) |

Table 21: **Encoder Ablation for Interventional Generation.** We compare Cond-FiP against the baselines for the task of generating interventional data from input noise variables against two variants. One variant corresponds to the encoder trained on SCMs with only linear functional relationships, Cond-FiP(LIN). Similarly, we have another variant where the decoder was trained on SCMs with only rff functional relationships, Cond-FiP(RFF). Each cell reports the mean (standard error) RMSE over the multiple test datasets for each scenario. *Results show that training on only non-linear SCMs* (Cond-FiP(RFF)) *gives similar performance as training on both linear and non-linear SCMs* (Cond-FiP).

 **G.2    Ablation Study of Decoder**

 We conduct an ablation study where we train two variants of the decoder Cond-FiP described as
 follows:

   • *Cond-FiP (LIN):* We sample SCMs with linear functional relationships during training.

   • *Cond-FiP (RFF):* We sample SCMs with non-linear functional relationships for training.

 Note that in the main results (Table 2, Table 3) we show the performances of Cond-FiP trained by
 sampling SCMs with both linear and non-linear causal mechanisms. Hence, this ablations helps
 us to understand whether the strategy of training on mixed causal mechanisms can bring more
 generalization to the amortization process, or if we should have trained decoders specialized for linear
 and non-linear functional relationships.

 We present the results of our ablation study in Table 22 and Table 23, for the task of sample generation
 and interventional generation respectively. Our findings indicate that Cond-FiP decoder trained
 for both linear and non-linear functional relationships is able to specialize for both the scenarios.
 While Cond-FiP (LIN) is only able to perform well for linear benchmarks, and similarly Cond-FiP
 (RFF) can only achieve decent predictions for non-linear benchmarks, Cond-FiP is achieve the best
 performances on both the linear and non-linear benchmarks.

| Method | Total Nodes | LIN **IN** | RFF **IN** | LIN **OUT** | RFF **OUT** |
|---|---|---|---|---|---|
| Cond-FiP(LIN) | 10 | 0.07 (0.02) | 0.4 (0.06) | 0.07 (0.01) | 0.25 (0.06) |
| Cond-FiP(RFF) | 10 | 0.1 (0.02) | 0.15 (0.02) | 0.08 (0.01) | 0.09 (0.01) |
| Cond-FiP | 10 | 0.06 (0.01) | 0.14 (0.02) | 0.05 (0.01) | 0.08 (0.01) |
| Cond-FiP(LIN) | 20 | 0.07 (0.01) | 0.44 (0.07) | 0.10 (0.01) | 0.58 (0.02) |
| Cond-FiP(RFF) | 20 | 0.11 (0.01) | 0.26 (0.06) | 0.14 (0.01) | 0.31 (0.03) |
| Cond-FiP | 20 | 0.05 (0.01) | 0.24 (0.06) | 0.07 (0.01) | 0.3 (0.03) |
| Cond-FiP(LIN) | 50 | 0.10 (0.01) | 0.5 (0.07) | 0.14 (0.02) | 0.69 (0.04) |
| Cond-FiP(RFF) | 50 | 0.15 (0.02) | 0.27 (0.05) | 0.19 (0.02) | 0.5 (0.07) |
| Cond-FiP | 50 | 0.08 (0.01) | 0.25 (0.05) | 0.07 (0.0) | 0.48 (0.07) |
| Cond-FiP(LIN) | 100 | 0.1 (0.01) | 0.51 (0.07) | 0.15 (0.02) | 0.72 (0.04) |
| Cond-FiP(RFF) | 100 | 0.16 (0.03) | 0.29 (0.07) | 0.27 (0.04) | 0.59 (0.06) |
| Cond-FiP | 100 | 0.07 (0.01) | 0.29 (0.07) | 0.09 (0.01) | 0.57 (0.07) |

Table 22: **Decoder Ablation for Sample Generation.** We compare Cond-FiP for the task of generating samples from input noise variables against two variants. One variant corresponds to a decoder trained on SCMs with only linear functional relationships, Cond-FiP(LIN). Similarly, we have another variant where the decoder was trained on SCMs with only rff functional relationships, Cond-FiP(RFF). Each cell reports the mean (standard error) RMSE over the multiple test datasets for each scenario. *Results indicate that training on both linear and non-linear SCMs is crucial to generalize effectively in all scenarios.*

| Method | Total Nodes | LIN **IN** | RFF **IN** | LIN **OUT** | RFF **OUT** |
|---|---|---|---|---|---|
| Cond-FiP(LIN) | 10 | 0.09 (0.02) | 0.40 (0.07) | 0.06 (0.01) | 0.22 (0.04) |
| Cond-FiP(RFF) | 10 | 0.16 (0.05) | 0.22 (0.03) | 0.08 (0.01) | 0.11 (0.01) |
| Cond-FiP | 10 | 0.10 (0.03) | 0.21 (0.03) | 0.07 (0.01) | 0.11 (0.01) |
| Cond-FiP(LIN) | 20 | 0.10 (0.01) | 0.45 (0.07) | 0.16 (0.03) | 0.57 (0.02) |
| Cond-FiP(RFF) | 20 | 0.14 (0.02) | 0.26 (0.05) | 0.21 (0.03) | 0.32 (0.02) |
| Cond-FiP | 20 | 0.09 (0.01) | 0.24 (0.05) | 0.14 (0.03) | 0.31 (0.03) |
| Cond-FiP(LIN) | 50 | 0.14 (0.02) | 0.49 (0.07) | 0.14 (0.02) | 0.68 (0.04) |
| Cond-FiP(RFF) | 50 | 0.19 (0.03) | 0.28 (0.05) | 0.21 (0.03) | 0.49 (0.06) |
| Cond-FiP | 50 | 0.13 (0.02) | 0.27 (0.04) | 0.12 (0.02) | 0.48 (0.07) |
| Cond-FiP(LIN) | 100 | 0.12 (0.02) | 0.52 (0.07) | 0.18 (0.03) | 0.71 (0.04) |
| Cond-FiP(RFF) | 100 | 0.18 (0.03) | 0.32 (0.07) | 0.24 (0.04) | 0.59 (0.07) |
| Cond-FiP | 100 | 0.10 (0.01) | 0.30 (0.06) | 0.14 (0.02) | 0.58 (0.07) |

Table 23: **Decoder Ablation for Interventional Generation.** We compare Cond-FiP against two variants for the task of interventional data from input noise variables. One variant corresponds to a decoder trained on SCMs with only linear functional relationships, Cond-FiP(LIN). Similarly, we have another variant where the decoder was trained on SCMs with only rff functional relationships, Cond-FiP(RFF). Each cell reports the mean (standard error) RMSE over the multiple test datasets for each scenario. *Results indicate that training on both linear and non-linear SCMs is crucial to generalize effectively in all scenarios.*

## H  Experiments on CSuite with Complex Noise Distributions

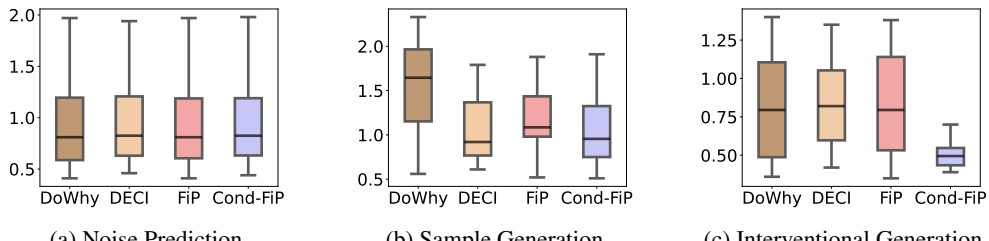

(a) Noise Prediction        (b) Sample Generation        (c) Interventional Generation

Figure 7: We compare Cond-FiP against the baselines for the different evaluation tasks on the **Large Backdoor** and **Weak Arrow** datasets from the **CSuite benchmark**, where the noise distribution is modified to be a multi-modal gaussian mixture model. We experiment with 6 different cases of the noise distribution for each dataset. The y-axis denotes the RMSE for the respective tasks across the 12 scenarios (datasets & noise distribution). *Results indicate that Cond-FiP can generalize to instances with more complex noise distributions like gaussian mixture models.*

To conduct more OOD evaluations, we modify the noise distribution of the Large Backdoor and Weak Arrow datasets from the Csuite benchmark such that the noise variables are sampled from a guassian mixture model (GMM). We considered the following cases for the GMM noise distribution.

- Noise is sampled with equal probability from either $N(-2, 1)$ and $N(2, 1)$.
- Noise is sampled with equal probability from either $N(-2, 2)$ and $N(2, 2)$.
- Noise is sampled with equal probability from either $N(-2, 1)$ and $N(2, 2)$.
- Noise is sampled with equal probability from either $N(-5, 1)$ and $N(5, 1)$.
- Noise is sampled with equal probability from either $N(-5, 2)$ and $N(5, 2)$.
- Noise is sampled with equal probability from either $N(-5, 1)$ and $N(5, 2)$.

This leads to a total of 12 experimental setting with 6 different GMM noise distribution for both the Large Backdoor and Weak Arrow datasets from the CSuite benchmark. Results in Figure 7 demonstrate that Cond-FiP remains competitive with baselines across all tasks. Importantly, while baselines were trained from scratch for each specific gaussian mixture noise distribution, Cond-FiP was pretrained only on gaussian noise and generalizes effectively to settings with GMM noise distribtion.

# I  Experiments on Real World Benchmark

| Method | MMD($\widehat{D_{\boldsymbol{X}}^{\text{query}}}, D_{\boldsymbol{X}}^{\text{query}}$) | MMD($\widehat{D_{\boldsymbol{X}}^{\text{context}}}, D_{\boldsymbol{X}}^{\text{query}}$) | MMD($D_{\boldsymbol{X}}^{\text{context}}, D_{\boldsymbol{X}}^{\text{query}}$) |
|---|---|---|---|
| DoWhy | 0.015 | 0.014 | 0.005 |
| DECI | 0.014 | 0.005 | 0.005 |
| FiP | 0.015 | 0.005 | 0.005 |
| Cond-FiP | 0.013 | 0.005 | 0.005 |

Table 24: **Results for Sachs dataset.** We benchmark Cond-FiP against the baselines for the task of generating observational data on the real world Sachs benchmark. Each cell reports the MMD, and we also report the reconstruction error for all of the methods. *Results indicate that Cond-FiP matches the performance of baselines trained from scratch.*

We use the real world flow cytometry dataset [Sachs et al., 2005] to benchmark Cond-FiP againts the baselines. This dataset contains $n \simeq 800$ observational samples expressed in a $d = 11$ dimensional space, and the reference (true) causal graph. We split this into context $D_{\boldsymbol{X}}^{\text{context}} \in \mathbb{R}^{n_{\text{context}} \times d}$ and queries $D_{\boldsymbol{X}}^{\text{query}} \in \mathbb{R}^{n_{\text{query}} \times d}$, each of size $n_{\text{context}} = n_{\text{query}} = 400$. Note that the context dataset is to used to train the baselines and obtain dataset embedding for Cond-FiP, while the query dataset is used for evaluation of all the methods.

Since we don't have access to the true causal mechanisms, we cannot compute RMSE for noise prediction or sample generation like we did in our experiments with synthetic benchmarks. Instead for each method, we obtain the noise predictions $\widehat{D_{\boldsymbol{N}}^{\text{context}}}$ on the context, and use it to fit a gaussian distribution for each component (node). Then we use the learned gaussian distribution to sample new noise variables, $\widehat{D_{\boldsymbol{N}}^{\text{query}}}$, which are mapped to the observations as per the causal mechanisms learned by each method, $\widehat{D_{\boldsymbol{X}}^{\text{query}}}$. Finally, we compute the maximum mean discrepancy (MMD) distance between $\widehat{D_{\boldsymbol{X}}^{\text{query}}}$ and $D_{\boldsymbol{X}}^{\text{query}}$ as metric to determine whether the method has captured the true causal mechanisms. For consistency, we also evaluate the reconstruction performances of the models by using directly the inferred noise from context $\widehat{D_{\boldsymbol{N}}^{\text{context}}}$ from the models, and then compute MMD between their reconstructed data ($\widehat{D_{\boldsymbol{X}}^{\text{context}}}$) and the query data ($D_{\boldsymbol{X}}^{\text{query}}$).

Table 24 presents our results, where for reference we also report the MMD distance between samples from the context and query split, which should serve as the gold standard since both the datasets are sampled from the same distribution. We find that Cond-FiP is competitive with the baselines that were trained from scratch. Except DoWhy, the MMD distance with reconstructed samples from the methods are close to oracle performance.

**No Interventional Generation Results.** Note that Cond-FiP (and the other baselines considered in this work) only supports hard interventions while the interventional data available for Sachs are soft interventions (i.e. the interventional operations applied are unknown). Hence, we are unable to provide a comprehensive evaluation of Cond-FiP (as well as the other baselines) for interventional predictions on Sachs.

## J  Comparing Cond-FiP with CausalNF

We also compare Cond-FiP with CausalNF [Javaloy et al., 2023] for the task of noise prediction (Table 25) and sample generation (Table 26). The test datasets consist of $n_{\text{test}} = 400$ samples, exact same setup as in our main results (Table 1, Table 2, and Table 3). To ensure a fair comparison, we provided CausalNF with the true causal graph.

Our analysis reveals that CausalNF underperforms compared to Cond-FiP in both tasks, and it is also a weaker baseline relative to FiP. Note also the authors did not experiment with large graphs for CausalNF; the largest graph they used contained approximately 10 nodes. Also, they trained CausalNF on much larger datasets with a sample size of 20k, while our setup has datasets with 400 samples only.

| Method | Total Nodes | LIN **IN** | RFF **IN** | LIN **OUT** | RFF **OUT** |
|---|---|---|---|---|---|
| CausalNF | 10 | 0.16 (0.02) | 0.41 (0.09) | 0.38 (0.04) | 0.35 (0.02) |
| Cond-FiP | 10 | 0.06 (0.01) | 0.10 (0.01) | 0.07 (0.01) | 0.10 (0.01) |
| CausalNF | 20 | 0.18 (0.03) | 0.45 (0.12) | 0.29 (0.05) | 0.36 (0.03) |
| Cond-FiP | 20 | 0.06 (0.01) | 0.09 (0.01) | 0.07 (0.00) | 0.12 (0.00) |
| CausalNF | 50 | 0.25 (0.03) | 0.56 (0.09) | 0.45 (0.06) | 0.38 (0.04) |
| Cond-FiP | 50 | 0.06 (0.01) | 0.10 (0.01) | 0.07 (0.01) | 0.14 (0.01) |
| CausalNF | 100 | 0.24 (0.02) | 0.80 (0.1) | 0.37 (0.06) | 0.49 (0.05) |
| Cond-FiP | 100 | 0.05 (0.0) | 0.10 (0.01) | 0.07 (0.01) | 0.16 (0.01) |

Table 25: **Results for Noise Prediction with CausalNF.** We compare Cond-FiP against CausalNF for the task of predicting noise variables from input observations. *We find that CausalNF underperforms compared to Cond-FiP by a significant margin.*

| Method | Total Nodes | LIN **IN** | RFF **IN** | LIN **OUT** | RFF **OUT** |
|---|---|---|---|---|---|
| CausalNF | 10 | 0.27 (0.07) | 0.29 (0.04) | 0.20 (0.03) | 0.20 (0.03) |
| Cond-FiP | 10 | 0.06 (0.01) | 0.14 (0.02) | 0.05 (0.01) | 0.08 (0.01) |
| CausalNF | 20 | 0.23 (0.02) | 0.36 (0.05) | 0.22 (0.02) | 0.45 (0.02) |
| Cond-FiP | 20 | 0.05 (0.01) | 0.24 (0.06) | 0.07 (0.01) | 0.30 (0.03) |
| CausalNF | 50 | 1.5 (0.26) | 0.93 (0.13) | 3.09 (0.55) | 0.95 (0.04) |
| Cond-FiP | 50 | 0.08 (0.01) | 0.25 (0.05) | 0.07 (0.00) | 0.48 (0.07) |
| CausalNF | 100 | 1.23 (0.13) | 0.85 (0.08) | 1.67 (0.13) | 0.96 (0.04) |
| Cond-FiP | 100 | 0.07 (0.01) | 0.29 (0.07) | 0.09 (0.01) | 0.57 (0.07) |

Table 26: **Results for Sample Generation with CausalNF.** We compare Cond-FiP against CausalNF for the task of generating samples from input noise variables. *We find that CausalNF underperforms compared to Cond-FiP by a significant margin.*

## K  Limitations of Cond-FiP

### K.1  Evaluating Generalization of Cond-Fip to Larger Sample Size

In the main results (Table 1, Table 2, and Table 3), we evaluated Cond-FiP's generalization capabilities to larger graphs ($d = 50$, $d = 100$) than those used for training ($d = 20$). In this section, we carry a similar experiment where instead of increasing the total nodes in the graph, we test Cond-FiP on datasets with more samples $n_{\mathcal{D}_{\text{test}}} = 1000$, while Cond-FiP was only trained for datasets with sample size $n_{\mathcal{D}} = 400$.

The results for the experiments are presented in Table 27, Table 28, and Table 29 for the task of noise prediction, sample generation, and interventional generation respectively. Our findings indicate that Cond-FiP is still able to compete with other baseline in this regime. However, we observe that the performances of Cond-FiP did not improve by increasing the sample size compared to the results obtained for the 400 samples case, meaning that the performance of our models depends exclusively on the setting used at training time. We leave for future works the learning of a larger instance of Cond-FiP trained on larger sample size problems.

| Method | Total Nodes | LIN **IN** | RFF **IN** | LIN **OUT** | RFF **OUT** |
|---|---|---|---|---|---|
| DoWhy | 10 | 0.02 (0.0) | 0.10 (0.01) | 0.21 (0.04) | 0.23 (0.02) |
| DECI | 10 | 0.05 (0.01) | 0.12 (0.01) | 0.21 (0.04) | 0.27 (0.03) |
| FiP | 10 | 0.03 (0.0) | 0.06 (0.0) | 0.21 (0.04) | 0.23 (0.02) |
| Cond-FiP | 10 | 0.05 (0.01) | 0.11 (0.01) | 0.21 (0.04) | 0.25 (0.02) |
| DoWhy | 20 | 0.02 (0.0) | 0.11 (0.02) | 0.16 (0.01) | 0.3 (0.02) |
| DECI | 20 | 0.04 (0.01) | 0.11 (0.02) | 0.16 (0.01) | 0.29 (0.02) |
| FiP | 20 | 0.03 (0.0) | 0.08 (0.02) | 0.16 (0.01) | 0.26 (0.02) |
| Cond-FiP | 20 | 0.06 (0.01) | 0.09 (0.01) | 0.18 (0.01) | 0.26 (0.01) |

Table 27: **Results for Noise Prediction with Larger Sample Size ($n_{\mathcal{D}_{\text{test}}} = 1000$).** We compare Cond-FiP against the baselines for the task of predicting noise variables from the input observations. Each cell reports the mean (standard error) RMSE over the multiple test datasets for each scenario. *Results indicate that Cond-FiP does not yet benefit from larger context sizes at inference, suggesting the need to scale both the model and training data for richer contexts.*

| Method | Total Nodes | LIN **IN** | RFF **IN** | LIN **OUT** | RFF **OUT** |
|---|---|---|---|---|---|
| DoWhy | 10 | 0.04 (0.0) | 0.14 (0.02) | 0.29 (0.04) | 0.3 (0.03) |
| DECI | 10 | 0.07 (0.01) | 0.17 (0.02) | 0.29 (0.04) | 0.33 (0.04) |
| FiP | 10 | 0.05 (0.0) | 0.09 (0.01) | 0.29 (0.04) | 0.29 (0.03) |
| Cond-FiP | 10 | 0.05 (0.01) | 0.14 (0.02) | 0.29 (0.04) | 0.29 (0.03) |
| DoWhy | 20 | 0.04 (0.01) | 0.21 (0.05) | 0.28 (0.01) | 0.55 (0.06) |
| DECI | 20 | 0.07 (0.01) | 0.21 (0.04) | 0.29 (0.01) | 0.59 (0.06) |
| FiP | 20 | 0.05 (0.0) | 0.17 (0.04) | 0.28 (0.01) | 0.53 (0.06) |
| Cond-FiP | 20 | 0.05 (0.0) | 0.24 (0.05) | 0.28 (0.01) | 0.53 (0.06) |

Table 28: **Results for Sample Generation with Larger Sample Size** ($n_{\mathcal{D}_{\text{test}}} = 1000$). We compare Cond-FiP against the baselines for the task of generating samples from the input noise variables. Each cell reports the mean (standard error) RMSE over the multiple test datasets for each scenario. *Results indicate that Cond-FiP does not yet benefit from larger context sizes at inference, suggesting the need to scale both the model and training data for richer contexts.*

| Method | Total Nodes | LIN **IN** | RFF **IN** | LIN **OUT** | RFF **OUT** |
|---|---|---|---|---|---|
| DoWhy | 10 | 0.04 (0.01) | 0.16 (0.03) | 0.26 (0.03) | 0.27 (0.03) |
| DECI | 10 | 0.09 (0.01) | 0.19 (0.02) | 0.26 (0.03) | 0.31 (0.04) |
| FiP | 10 | 0.05 (0.01) | 0.12 (0.02) | 0.26 (0.03) | 0.27 (0.03) |
| Cond-FiP | 10 | 0.09 (0.02) | 0.19 (0.03) | 0.27 (0.03) | 0.3 (0.03) |
| DoWhy | 20 | 0.04 (0.0) | 0.20 (0.04) | 0.26 (0.01) | 0.53 (0.06) |
| DECI | 20 | 0.08 (0.01) | 0.20 (0.03) | 0.29 (0.02) | 0.54 (0.05) |
| FiP | 20 | 0.06 (0.01) | 0.16 (0.04) | 0.28 (0.02) | 0.48 (0.06) |
| Cond-FiP | 20 | 0.07 (0.01) | 0.27 (0.05) | 0.30 (0.02) | 0.51 (0.06) |

Table 29: **Results for Interventional Generation with Larger Sample Size** ($n_{\mathcal{D}_{\text{test}}} = 1000$). We compare Cond-FiP against the baselines for the task of generating interventional data from the input noise variables. Each cell reports the mean (standard error) RMSE over the multiple test datasets for each scenario. *Results indicate that Cond-FiP does not yet benefit from larger context sizes at inference, suggesting the need to scale both the model and training data for richer contexts.*

 **K.2  Counterfactual Generation with Cond-FiP**

We provide results (Table 30) for bechmarking Cond-FiP against baselines for the task of counter-factual generation. We operate in the same setup as the one in our main results ($n_{\mathcal{D}_{\text{test}}} = 400$) Appendix C and all the methods are provided with the true casual graph. We observe that Unlike the tasks of noise prediction, sample & interventional generation, we find that Cond-FiP is worse than the baselines for the task of counterfactual generation. This can be explained as the training of Cond-FiP decoder relies on the true noise variables, and the model struggles to generalize the learned functional mechanisms when provided with inferred noise variables. We leave the improvement of Cond-FiP for counterfactual generation as future work.

| Method | Total Nodes | LIN **IN** | RFF **IN** | LIN **OUT** | RFF **OUT** |
|---|---|---|---|---|---|
| DoWhy | 10 | 0.03 (0.03) | 0.13 (0.03) | 0.0 (0.0) | 0.04 (0.01) |
| DECI | 10 | 0.1 (0.02) | 0.2 (0.03) | 0.04 (0.01) | 0.11 (0.02) |
| FiP | 10 | 0.03 (0.01) | 0.09 (0.02) | 0.02 (0.0) | 0.03 (0.01) |
| Cond-FiP | 10 | 0.09 (0.03) | 0.21 (0.03) | 0.05 (0.01) | 0.11 (0.01) |
| DoWhy | 20 | 0.01 (0.0) | 0.12 (0.03) | 0.0 (0.0) | 0.13 (0.02) |
| DECI | 20 | 0.06 (0.01) | 0.15 (0.03) | 0.07 (0.03) | 0.15 (0.02) |
| FiP | 20 | 0.03 (0.01) | 0.1 (0.03) | 0.06 (0.04) | 0.09 (0.02) |
| Cond-FiP | 20 | 0.09 (0.02) | 0.26 (0.05) | 0.13 (0.02) | 0.3 (0.03) |
| DoWhy | 50 | 0.0 (0.0) | 0.09 (0.02) | 0.0 (0.0) | 0.17 (0.04) |
| DECI | 50 | 0.04 (0.01) | 0.11 (0.02) | 0.03 (0.01) | 0.18 (0.04) |
| FiP | 50 | 0.03 (0.01) | 0.08 (0.02) | 0.03 (0.01) | 0.14 (0.04) |
| Cond-FiP | 50 | 0.1 (0.02) | 0.26 (0.04) | 0.1 (0.01) | 0.46 (0.06) |
| DoWhy | 100 | 0.0 (0.0) | 0.08 (0.02) | 0.0 (0.0) | 0.2 (0.05) |
| DECI | 100 | 0.02 (0.01) | 0.1 (0.02) | 0.02 (0.01) | 0.22 (0.05) |
| FiP | 100 | 0.01 (0.01) | 0.07 (0.02) | 0.02 (0.01) | 0.19 (0.05) |
| Cond-FiP | 100 | 0.09 (0.02) | 0.29 (0.06) | 0.13 (0.02) | 0.56 (0.08) |

Table 30: **Results for Counterfactual Generation.** We compare Cond-FiP against the baselines for the task of generating counterfactual data from the input noise variables. Each cell reports the mean (standard error) RMSE over the multiple test datasets for each scenario. Shaded rows denote the case where the graph size is larger than the train graph sizes ($d = 20$) for Cond-FiP. *Results indicate that Cond-FiP struggles with counterfactual generation and cannot always match the performance of baselines trained from scratch.*

# L  Broader Impact

We propose novel methodology for amortized inference of causal mechanisms in structural causal models, representing an initial step toward the development of causal foundational models. Integrating causal principles into machine learning has been widely suggested to improve robustness and reliability, an important property for high-stakes domains such as healthcare, policy, and scientific discovery. By advancing core methodology in causal inference, our work may indirectly support the creation of machine learning systems that are more transparent and trustworthy. However, our research currently does not target any societal application, and does not pose foreseeable risks or negative consequences.

