# OpenReview forum: "Amortized Inference of Causal Models via Conditional Fixed-Point Iterations"
_NeurIPS.cc/2025/Conference — Submitted to NeurIPS 2025_

### Official Review · Reviewer_cobb · 2025-06-23

**Clarity:** 3
**Significance:** 2
**Originality:** 2
**Rating:** 4
**Confidence:** 3

**Summary:**

The paper proposed a conditional fixed point iteration algorithm to allow flexible SCM generation from different observational distributions via conditioning on dataset embeddings. The method can generation observation and interventional data from novel SCMs at inference time.

**Questions:**

See above.
1. How many dataset size the data encoder could accomodate in terms of number of samples and number of nodes? How does that compare to similar existing methods, e.g. TabPFN?

Happy to reconsider the scores if the authors address concerns above.

**Ethical Concerns:**

["NO or VERY MINOR ethics concerns only"]

**Final Justification:**

The authors have addressed most of my concerns from noise variables during training to direct extraction of causal mechanisms, computational efficiency and scalability. Things to improve are the reliance on other causal graph discovery methods like AVICI that result to high OOD error, and lack of interpretability on learned mechanisms. Despite potential improvement points for future direction, the paper itself already serves an interesting contribution to the community. Therefore I maintain my score and leaning towards acceptance.

**Limitations:**

Yes.

**Paper Formatting Concerns:**

No issue identified.

**Quality:**

3

**Strengths And Weaknesses:**

Strengths:
- propose a new FiP based approach for SCM generation (both observational and interventional) that accomodates different distributions (a meta-learning extension to original Fip approach)

Weaknesses:
- not completely realistic assumption to know G in the inference time, even though it could be estimated (performed experiments in Fig 5). The OOD error though is best among methods compared it is still very big to be of realistic usage.
- Assumption of ANM - may be restrictive in realistic settings
- No discussion or experiments on real world applications. Unclear the usage of proposed method. Could the authors discuss or demonstrate the real world applications? e.g., why knowing noise (z) is a good assumption to start generation? Often real world is given observed dataset to estimate causal effects or structure discovery and noise is often unobtainable / ignored.
- The novelty is around direct extraction of causal mechanisms through $F(z)$. Unclear demonstrations of the exact advantages in direct extraction of causal mechanisms without noise. One advantage could be interpretability or understanding of each individual causal mechanisms. However, so far as deep-learning approach, I do not see a clear path forward for mechanism disentanglement. Could the authors discuss the particular advantages for direct extraction of causal mechanisms without noise?
- The method focuses on advantages in sample generation for novel SCMs. Often sampling method prefers computational efficiency. Could the authors compare time taken to train, and sample with baseline methods / comparable methods to demonstrate if there is computational efficiency advantage in the proposed method.

---

> ### Author Rebuttal · Authors · 2025-07-31
>
> We thank the reviewer for their positive and insightful feedback! We now address the reviewer’s concerns below.
>
> ### **Real-World Applications**
> ---
> We agree with the reviewer that Cond-FiP cannot be trained on real-world dataset as we don't have access to their noise variables. However, it is important to note that during inference, Cond-FiP does not require access to noise samples. Instead, it only needs the observations (and a predicted or true causal graph) to infer the functional mechanisms. This allows Cond-FiP to be applied to practical real-world datasets, as demonstrated in our experiment on the Sachs dataset (Appendix I), mentioned in lines 285-288 in the main paper.
>
> ### **Clarifications regarding ANM assumption and noise variables during training**
> ---
> While the ANM assumption may be seen as a limitation, we would like to clarify that our method relies on the ANM assumption only for training the encoder. This is because we need the encoder to predict the noise from the data in order to obtain embeddings, which is simplified under the ANM assumption, as explained in Appendix A.2.
>
> However, as stated in lines 318–322, noise variables are only used as targets to train the encoder (hence the ANM assumption), and are not necessary for training the decoder. In principle, the proposed encoder-decoder architecture supports end-to-end training via reconstruction loss alone, without access to noise variables. This end-to-end approach shows good in-distribution and OOD generalization for linear functional relationships. But it struggled in the more complex non-linear (RFF) case, motivating our use of noise variable recovery as a "pretraining" step for the encoder.
>
> *Future Directions for Training with Real-World Datasets:* An interesting direction for incorporating realistic SCMs during training is to develop more general encoder training strategies, such as using self-supervised learning for dataset encoding. Another option is to pursue end-to-end training of Cond-FiP, or adopt curriculum learning by first training the encoder to predict  noise variables, and then fine-tuning it using the decoder's reconstruction loss on more realistic SCMs. However, we consider these extensions beyond the scope of the current work.
>
> Finally, we emphasize that while prior work on amortized causal learning has largely focused on causal effect estimation or causal discovery (as discussed in Section 5), our work tackles the novel task of amortized inference of causal mechanisms of SCMs. *We view our experiments with additive noise models as a critical proof-of-concept that demonstrates the feasibility and potential of this direction, supported by extensive evaluations and ablation studies validating our claims.*  We believe our framework can serve as a good motivation for future works that incorporate real-world datasets during training.
>
> ### **Regarding OOD error in the experiment without true causal graph**
> ---
> We agree with the reviewer that assuming access to the true causal graph at inference time is a strong and often unrealistic assumption. To address this, we conducted additional experiments (Fig. 5) where we first inferred the graph using AVICI, and then performed inference using this estimated graph with both Cond-FiP and the baselines.
>
> Notably, although Cond-FiP was trained only with instances containing the true causal graph, it generalizes as well as the baselines that were trained from scratch using the inferred (and potentially incorrect) graphs. This provides more evidence to our claim that Cond-FiP learns causal mechanisms during training and can adapt to new contexts at test time by inferring functions that best explain the available information in the context (input graph and observations), even when the input causal graph is inaccurate.
>
> Regarding the absolute OOD error, this could indicate limitations in the quality of the inferred graph from AVICI. Especially since the baselines that were trained from scratch also obtain high error, it might be the case that inferred causal graphs may not capture the true causal graph well enough to support reliable inference of causal mechanisms from observed data.
>
> ### **Regarding direct extraction of causal mechanisms via $F(z)$**
> ---
> We are not entirely certain we have fully understood the concern, and it would greatly appreciate more feedback. If the question is whether training the decoder requires access to the noise-free mechanism output $F(z)$ versus the noisy target $z= F(z) + n$, then we would like to clarify that this distinction does not affect the learning problem in our setup.
>
> In line with the original FiP paper, the objective of our conditional variant of FiP is to predict the observed variable $z_i$ from its parents $p_i$, hence learn the underlying mechanism $F_i(p_i)$, where $F= [F_1(p_1), \cdots, F_d(p_d)]$. Since the noise $n_i$ is independent of $F_i(p_i)$, learning the mechanism  $F_i(p_i)$ from the noisy target ($z_i$) versus the noise-free target ($z_i - n_i$) does change the result of optimization.
>
> Regarding the interpretability or disentanglement of the mechanisms, since our method relies on deep learning, the learned functions are not directly interpretable. However, we agree this is an important and promising direction for future work.
>
> Please let us know if this addresses your concern. If we have misunderstood the question, we would be happy to engage in further discussion.
>
> ### **Computation efficiency of Cond-FiP versus baselines at inference**
> ---
> Like any typical amortized approach, Cond-FIP incurs a higher training cost compared to baselines, as it is trained across a large number of tasks. While each forward pass in Cond-FIP has a computational cost comparable to that of FIP, the amortized training involves approximately 4M datasets. However, this cost is offset by a significant efficiency gain at inference time as Cond-FIP requires only a single forward pass to make predictions for a novel task, whereas baselines like FIP must be retrained from scratch for each task.
>
> In practice, Cond-FIP can infer causal mechanisms for a novel task in under one minute, while FIP takes on average 30 minutes per task. For a concrete comparison, it took us 30 hours to train Cond-FIP but we can solve each inference task in max 1 minute. Therefore, to compute our main results (Tables 1–3) we evaluated 360 different tasks, implying a total cost of $30 + 360/60 = 36$ hours with Cond-FiP.  In contrast, evaluating FIP on the same 360 tasks would require retraining from scratch each time, taking approximately $180$ hours!
>
> Thus, while Cond-FIP has a higher one-time training cost, it offers a *5× speedup* over FiP in total runtime when evaluating across multiple tasks.
>
> ### **Scalability of Cond-FiP**
> ---
> Thank you for raising this point. The transformer architecture we use is indeed scalable in the same way standard transformers (TabPFN) with non-flash attention are. Given our compute budget (a single L40 GPU per experiment), we currently train on graphs with $d=20$ nodes and $n=400$ samples. However, we have verified that the model can scale to $n=2000$ samples when keeping $d=20$ fixed, and up to $d=400$ nodes when keeping $n=400$ fixed.
>
> It's important to note that the primary bottleneck in scaling to larger graphs (during training) is not due to the transformer model itself, but rather the cost of data generation. Our training pipeline involves generating data from randomly sampled SCMs at each step, and this process becomes significantly slower as the graph size increases.
>
> ---
> Thank you once again for your constructive comments! We are open to further discussion and would be happy to address any remaining concerns.

---

> > ### Comment · Reviewer_cobb · 2025-08-04
> >
> > Thank you for the authors on detailed response and it is interesting to read. It addressed most of my concerns, I am keeping my score and leaning towards acceptance.

---

> ### Author Response · Authors · 2025-08-04
>
> We appreciate that you found our response interesting and it is encouraging to know that most of your concerns have been addressed. We remain available for any further discussion or clarification you may need, and we would be grateful if you would consider increasing your score.

---

### Official Review · Reviewer_m6sH · 2025-06-26

**Clarity:** 4
**Significance:** 4
**Originality:** 4
**Rating:** 5
**Confidence:** 4

**Summary:**

Cond-FiP is a method for what I would call causal estimation, not causal discovery. It assumes the data D and the underlying DAG G are both given and tries to estimate the functional forms of each variable given its parents in G, by a training phase that iterates over millions of additive noise models where noises and functions are known (and where masking can occur based on DAG structure) that results in a trained encoder. Then it uses this in a decoding phase, given a new dataset D' and DAG G', again with masking based on the DAG structure of G', to estimate the functions (with now unknown noise distributions) of each node in G' given its parents in G', again by applying masking.

The authors compare Cond-FiP to other methods of causal estimation (like DoWhy) for in-sample and out-of-sample estimation of these functions, and also look at scarce data regimes.

**Questions:**

1. Have you considered offering a whole-model estimation statistic that could be used to decide between different estimates of G' in the inference step?

2. Is there a way to sidestep needing to know the true noise distributions during training?

3. Is there a way to infer counterfactuals here that perhaps didn't occur to me?

4. Could one give a practical upper bound on the size of the graph G' that can be estimated in the inference phase, given that the training has all been done on 20-node SCMs?

5. Is it useful to try to infer explicit functions?

6. I thought I saw "scarce data" being written as "scare data" somewhere; perhaps this can be checked.

7. What *is* the density of these graphs? One imagines that if the graphs are very sparse (say, #edges = #nodes or #edges = 2 * #nodes) the problem might be considerably easier than otherwise.

**Ethical Concerns:**

["NO or VERY MINOR ethics concerns only"]

**Final Justification:**

The authors have made several improvements to readability, which was my main concern, and I am happy with these.

**Limitations:**

The authors themselves identified some of the questions raised above as limitations; I would simply like to know whether they've given further consideration to them. No societal impacts were suggested.

**Paper Formatting Concerns:**

I did not notice any formatting issues.

**Quality:**

4

**Strengths And Weaknesses:**

It took considerable sleuthing to decipher what was happening in this paper. It may help if, in the abstract, it were stated that the goal is to take a dataset and a known DAG and to estimate the functions in a given test SCM based on known example SCMs where the truth is known and noise distributions in particular are known. Additionally, it would be helpful to specify the density of the graphs in training and testing; if these are particularly sparse, the usefulness of the method may be questioned. That said, given this is the goal, and assuming the graphs are sufficiently complex, this is a well-thought-out method for performing causal estimation within the encoder-decoder paradigm. The experimental results show promise. It is very helpful that a single model is learned in the training phase, which can then be applied many times in the inference phase. It is helpful that large graphs can be estimated during the inference phase, despite smaller graphs being used for training. It is helpful that the learned graphs are sufficiently accurate to simulate interventions and that the scarce data regime has been considered.

The most obvious weakness of the approach is that it relies on another method for causal discovery in the inference phase, and thus is at the mercy of the user's chosen method, as this has not been experimentally validated in this context. This is a problem with DoWhy and other causal estimation frameworks as well. There is a tendency to say that one should rely on expert judgment for finding the G for inference. Still, in truth, that is rarely satisfactory, especially for the 100-node graphs that are being considered here for the inference phase. It is not entirely fair to treat this as a separate, prior step to estimation, as we know from linear estimation that obtaining a good estimate of the causal structure can enhance the accuracy of functional estimation. In fact, if an overall measure of fitness were available (as with the model chi-square in the linear case), one would see that different estimated causal graphs (estimated by an ideal procedure) may well have different fitnesses. If this problem is treated as completely antecedent to the current estimation procedure, a proper response might be to ask why an overall measure of fitness has not been offered so that estimations from different choices of G' can be compared for overall fitness. But this put causal discovery squarely back into the realm of causal estimation: one wants to find a graph G* that maximizes the fit of the noises and functions after estimation.

Another weakness is that noise distributions must be known during the training phase, which prevents training on real-world data.

Additionally, although interventions are shown to be supported, it is unclear whether counterfactuals can be inferred.

It is unclear whether the transformer architecture being used here is scalable to large datasets or complex graphs.

A weakness that could be addressed in future work is to infer explicit functional relationships from the decoded function using, for example, symbolic regression.

---

> ### Author Rebuttal · Authors · 2025-07-31
>
> We thank the reviewer for their insightful feedback! We appreciate that they found our experimental results promising, especially the generalization of Cond-FiP to larger graphs at inference than used in training, and superior generalization in the scarce data regime.
>
> We now address the reviewer’s concerns below.
>
> ### **Density of graphs**
> ---
> We clarify that Cond-FiP is trained and evaluated on graphs with varying levels of density, $\rho \in \{2, 3, 4\}$, where $\rho$ denotes the average number of edges per node. Hence, our results include both sparse and dense graphs, including cases ($\rho=3$, $\rho=4$) where the number of edges exceeds twice the number of nodes.
>
> ### **Deciding between different graphs at inference**
> ---
> Thank you for the thoughtful comment. We agree that treating causal discovery as entirely separate from estimation can be problematic, especially in large-scale settings where expert knowledge may not be sufficient. To address this, we use the reconstruction loss (RMSE) for the sample generation task as a whole-model fit criterion to compare candidate graphs $G^{'}$ during inference.
>
> We validate this approach by the following experiment where we create multiple candidate graphs by introducing random perturbations ($p$) to the true causal graph. Specifically, we randomly remove a proportion $p$ of the true edges, such that on average $𝑝 \times$  (total edges) are missing.  Due to space constraints, we report results for sample generation in the challenging OOD setting (d=100 & RFF OUT).
>
>
> |          | p=0 | p=0.01 | p=0.02 | p=0.05 | p=0.1 |
> |----------|--------|--------|--------|--------|-------|
> | FiP      | 0.55 (0.08)  | 0.55 (0.08)     |  0.57 (0.08)      |   0.62 (0.08)     |   0.68 (0.08)    |
> | Cond-FiP | 0.57 (0.07)  | 0.58 (0.07)     |  0.59 (0.07)      |   0.62 (0.07)     |   0.67 (0.07)    |
>
> Our results show that the reconstruction loss degrades as more edges are removed for both FiP and Cond-FiP, indicating that RMSE can serve as a reliable proxy for model fit and guide graph selection.
>
> *Metric for Real-World Datasets:* Note that for real-world scenarios where ground-truth noise is unavailable, we can instead use the MMD metric between generated samples and the observed empirical distribution (as described in our real-data experiments; Appendix I) to assess the sample quality and overall model fit.
>
> ### **Clarification about access to noise variables during training**
> ---
> We agree with the reviewer that access to noise variables prevents us from training with real-world data. However, as stated in lines 318–322, noise variables are only used as targets to train the encoder (hence the ANM assumption), and are not necessary for training the decoder. In principle, the proposed encoder-decoder architecture supports end-to-end training via reconstruction loss alone, without access to noise variables. This end-to-end approach shows good in-distribution and OOD generalization for linear functional relationships. But it struggled in the more complex non-linear (RFF) case, motivating our use of noise variable recovery as a "pretraining" step for the encoder.
>
> An interesting direction for incorporating realistic SCMs during training is to develop more general encoder training strategies, such as using self-supervised learning for dataset encoding. Another option is to pursue end-to-end training of Cond-FiP, or adopt curriculum learning by first training the encoder to predict  noise variables, and then fine-tuning it using the decoder's reconstruction loss on more realistic SCMs. However, we consider these extensions beyond the scope of the current work.
>
> Finally, we emphasize that while prior work on amortized causal learning has largely focused on causal effect estimation or causal discovery (as discussed in Section 5), our work tackles the novel task of amortized inference of causal mechanisms of SCMs. *We view our experiments with additive noise models as a critical proof-of-concept that demonstrates the feasibility and potential of this direction, supported by extensive evaluations and ablation studies validating our claims.*  We believe our framework can serve as a good motivation for future works that incorporate real-world datasets during training.
>
> *Inference on Real-World Datasets:* We re-emphasize that during inference, Cond-FiP does not require access to noise variables. Instead, it only needs the observations (and a predicted or true causal graph) to infer the functional mechanisms. This enables its application to real-world datasets, as demonstrated in our Sachs experiment (Appendix I).
>
>
> ### **Computing counterfactuals with Cond-FiP**
> ---
> Yes, Cond-FiP is capable of generating counterfactual samples by first inferring the noise variables using the encoder and then generating interventional data conditioned on these inferred noise variables via the decoder. We had conducted experiments in Appendix K.1, where we compare Cond-FiP’s counterfactual predictions against various baselines. While Cond-FiP is slightly worse overall (as acknowledged in the limitations section, lines 325–327), it remains competitive in several cases. Improving counterfactual generation performance is an important direction for future work.
>
> ### **Scalability of Cond-FiP**
> ---
> Thank you for raising this point. The transformer architecture we use is indeed scalable in the same way standard transformers with non-flash attention are. Given our compute budget (a single L40 GPU per experiment), we currently train on graphs with $d=20$ nodes and $n=400$ samples. However, we have verified that the model can scale to $n=2000$ samples when keeping $d=20$ fixed, and up to $d=400$ nodes when keeping $n=400$ fixed.
>
> It's important to note that the primary bottleneck in scaling to larger graphs (during training) is not due to the transformer model itself, but rather the cost of data generation. Our training pipeline involves generating data from randomly sampled SCMs at each step, and this process becomes significantly slower as the graph size increases.
>
> ### **Upper bound on graph size at inference**
> ---
> Providing a theoretical bound on the largest graph to which the Cond-FiP can generalize is challenging. However, to obtain an empirical estimate, we evaluated performance on larger graphs with $d \in [125, 150, 175, 200 ]$. Due to space constraints, we report results for sample generation in the RFF IN setting.
>
> |          | d=125 | d=150 | d=175 | d=200 |
> |----------|--------|--------|--------|--------|
> | FiP      |  0.31 (0.04) | 0.34 (0.08)     |  0.37 (0.05)      |  0.43 (0.06)   |
> | Cond-FiP | 0.30 (0.05) |  0.33 (0.05)   |  0.37 (0.04)  |   0.42 (0.07)     |
>
> Our results show that Cond-FiP remains competitive with the FiP baseline trained from scratch in these scenarios. We note that estimating graphs with $d>200$ using only $n=400$ samples is likely infeasible even for the baselines. Therefore, we consider $d=200$ as a practical upper bound for our evaluations with a context length of $400$ samples.
>
> ### **Inferring explicit functions**
> ---
> We agree that inferring explicit functional relationships using symbolic regression would be valuable for improving interpretability. Since our method relies on deep learning, the learned functions cannot be easily interpreted, but this is indeed a promising direction for future work.
>
> ---
> We will also correct the typos ("scare data") you have mentioned in the updated version of our draft. Thank you once again for your constructive comments! We are open to further discussion and would be happy to address any remaining concerns.

---

> > ### Comment · Reviewer_m6sH · 2025-08-03
> >
> > I read these clarifications with interest and believe you have addressed all of my points, except for the clarity issue. I think that clarifying at the beginning of the paper that it is about estimating functions for a postulated causal graph, rather than about causal discovery itself, and making other similar comments to help "orient" the reader to what the paper is about, helps situate it for readers unfamiliar with the terrain. As I said, it took me a few trips through the paper to figure out exactly what was going on; comments like this at the beginning of the paper and even in the abstract would have reduced that to one pass for me.

---

> ### Author Response · Authors · 2025-08-03
>
> Thank you for your thoughtful response to our rebuttal, we are glad that it helped to address your concerns. Regarding the clarity issue you raised, we agree with your feedback and will incorporate the suggested improvements in the next revision of our draft. Since updates to the paper are not permitted during the rebuttal period, we outline the planned changes below.
>
> ### **Modified Abstract**
> ---
>
> Structural Causal Models (SCMs) offer a principled framework to reason about interventions and support out-of-distribution generalization, which are key goals in scientific discovery. However, the task of learning SCMs from observed data poses formidable challenges, and often requires training a separate model for each dataset. In this work, we propose an amortized inference framework that trains a *single* model to predict the causal mechanisms of SCMs conditioned on their observational data and corresponding causal graph. We first use a transformer-based architecture for amortized learning of dataset embeddings, and then extend the Fixed-Point Approach (FiP) to infer the causal mechanisms conditionally on their dataset embeddings. As a byproduct, our method can generate observational and interventional data from novel SCMs at inference time, without updating parameters. Empirical results show that our amortized procedure performs on par with baselines trained specifically for each dataset on both in and out-of-distribution problems, and also outperforms them in scare data regimes.
>
>
>
> ### **Modified Introduction**
> ---
>
> We change the third paragraph of our introduction as follows to highlight the goal of our paper in a better manner.
>
> In this work, we tackle the novel problem of amortized inference of causal mechanisms for additive noise SCMs. While prior research has primarily focused on amortized approaches for causal discovery or treatment effect estimation, our goal instead is to train a single model capable of inferring the causal mechanisms of novel SCMs, given their observational data and associated causal graph. We propose a two-step approach where we first learn dataset embeddings via in-context learning to represent the task-specific information. These embeddings are then used to condition the fixed-point (FiP) approach for modeling causal mechanisms. This conditional moification, termed *Cond-FiP*, enables the model to adapt the causal mechanism for each specific instance. Our key contributions are highlighted below.
>
> ---
> Finally, we would like to note that in our problem setup (Section 2.2), we clearly specify that our goal is to learn causal mechanisms, not perform causal discovery, with the input context comprising both the observed samples and the corresponding causal graph (Equation 3). That said, we agree that this distinction was not made sufficiently explicit in the abstract and introduction, and we hope that the proposed changes address this concern.
>
> Thank you once again for your constructive comments!  If you feel that your concerns have been fully addressed, we would greatly appreciate if you can increase your score accordingly.

---

> > ### Author Response · Authors · 2025-08-07
> >
> > As the discussion period is drawing to a close, we kindly ask if our latest comment has addressed your concern regarding the clarity issue. We have highlighted the proposed changes to the abstract and introduction, and we also plan to clarify in the problem setup section further by explicitly stating that, given the causal graph and an observed dataset generated from an SCM, our goal is to infer the functional mechanisms of the SCM.
> >
> > We would be happy to provide further clarifications should you have any additional questions. Thank you once again for your time and thoughtful feedback!

---

> > > ### Comment · Reviewer_m6sH · 2025-08-08
> > >
> > > I'm sorry, I got distracted this week. I will increase my score.

---

> > > > ### Author Response · Authors · 2025-08-09
> > > >
> > > > We sincerely appreciate the time you dedicated to reviewing our detailed response and additional experiments, and we are grateful for the score increase. We are glad our response was able to address your concerns.

---

### Official Review · Reviewer_Zi1b · 2025-06-29

**Clarity:** 3
**Significance:** 3
**Originality:** 3
**Rating:** 4
**Confidence:** 4

**Summary:**

This paper proposes Cond-FiP, an amortized inference framework for learning causal mechanisms in structural causal models (SCMs). Unlike prior methods that require training a new model for each dataset, Cond-FiP learns to generalize across SCMs by conditioning a fixed-point-based causal generative model (FiP) on dataset-specific embeddings. The model architecture includes a transformer-based dataset encoder that learns embeddings from observational data and known causal graphs, and a conditional FiP decoder that uses these embeddings to infer causal mechanisms. The trained model can then infer the causal mechanisms of a new, unseen SCM at inference time by being provided with a small sample of its observational data and its causal graph, without any additional training. Extensive experiments show that Cond-FiP performs competitively with models trained from scratch on each specific dataset, particularly in out-of-distribution scenarios and when data is scarce.

**Questions:**

1. Please see the comments and questions in the Weaknesses part.

2. While the authors evaluate performance with smaller context sizes during inference (50-100 observations), the training always uses 400 observations per SCM, which is large for many real-world domains. Could the authors investigate how Cond-FiP performs when trained on datasets with smaller sample sizes (50-100 observations) that better reflect real-world constraints?

3. Given that real-world applications often have uncertain or partially incorrect causal graphs, could the authors provide an analysis of Cond-FiP's robustness to causal graph inference errors?

**Ethical Concerns:**

["NO or VERY MINOR ethics concerns only"]

**Final Justification:**

I maintain my current score, leaning toward acceptance.

**Limitations:**

The authors adequately addressed the limitations and potential negative societal impact of their work in the Conclusion section and Appendix L.

**Paper Formatting Concerns:**

I do not find any formatting issues.

**Quality:**

3

**Strengths And Weaknesses:**

__Strengths__:

1. The paper addresses the important problem of amortized inference for causal mechanisms, which is essential for enabling zero-shot causal inference on unseen datasets (an important step towards building causal foundation models). The paper is clearly written and well-organized, making it easy to follow.

2. The integration of in-context learning with fixed-point causal modeling is both novel and well-justified. The proposed two-stage architecture, consisting of a transformer-based dataset encoder and a conditional FiP decoder, is technically sound.

3. The authors present thorough experimental evaluations across a wide range of settings. The results demonstrate that Cond-FiP performs competitively with baselines that require training from scratch on each dataset, often outperforming them in OOD and low-data regimes, demonstrating the success and the benefits of amortization.

__Weaknesses__:

1. __Limitation for realistic applications__: Cond-FiP requires access to true noise variables during training, which is only available in synthetic settings. Real-world data-generating processes are typically complex and cannot be fully parametrized using additive noise models. While realistic simulators (e.g., gene regulatory network simulators) could provide valuable domain knowledge for amortized learning, Cond-FiP cannot leverage such simulators since true noise variables are unavailable. This undermines one of the key motivations for amortized inference—handling implicit likelihoods in realistic scenarios. Training an amortized "foundation" model solely on synthetic SCMs can raise concerns about covering the complex relationships in scientific domains (SDEs, ODEs, etc.) and practical applicability to real-world scenarios.

2. __Insufficient evaluation for complex nonlinear mechanisms__: The authors only consider Random Fourier Features (RFF) for nonlinear relationships, which is inadequate for complex nonlinear functions. Multilayer perceptrons (MLPs) are more expressive and represent the standard approach for nonlinear function approximation in modern causal discovery [1]. It would be better if the authors also conduct experiments on MLP features in the nonlinear setting for evaluation.

3. __Inability to leverage interventional data__: The dataset encoder appears designed only for observational data and does not accommodate interventional inputs during training. As interventional data become increasingly accessible in scientific domains and their importance for causal identification beyond Markov equivalence classes, it would be great if the authors could discuss how to use interventional data in the Cond-FiP framework.

4. While Cond-FiP excels at noise prediction (the training objective), it shows degraded performance on sample generation and interventional generation compared to top baselines. Since these are key downstream use cases for causal models, the overall utility of Cond-FiP is somewhat diminished, despite its amortized learning advantages.

5. (Minor typos)

    (1) Page 1, line 13: "also outperforms them in scare data regimes" → should be "scarce data regimes"

    (2) Page 7, line 260: "Better Generalization in Scare Data Regimes" → should be "Scarce Data Regimes"

    (3) Page 9, line 312: "Cond-FiP not only generalizes to unseens in-distribution instances" → should be "unseen in-distribution instances"





[1] Lorch, Lars, et al. "Dibs: Differentiable Bayesian structure learning." Advances in Neural Information Processing Systems 34 (2021): 24111-24123.

---

> ### Author Rebuttal · Authors · 2025-07-31
>
> We thank the reviewer for their positive and insightful feedback! We are glad they like our setup of amortized inference of causal mechanisms, recognizing it as an important step toward building causal foundation models. We also appreciate that they found our approach novel and well-justified through comprehensive experiments.
>
> We now address the reviewer’s concerns below.
>
> ### **Clarifications about access to noise variables during training**
> ---
> We agree with the reviewer that access to noise variables prevents us from training with real-world data. However, as stated in lines 318–322, noise variables are only used as targets to train the encoder (hence the ANM assumption), and are not necessary for training the decoder. In principle, the proposed encoder-decoder architecture supports end-to-end training via reconstruction loss alone, without access to noise variables. This end-to-end approach shows good in-distribution and OOD generalization for linear functional relationships. But it struggled in the more complex non-linear (RFF) case, motivating our use of noise variable recovery as a "pretraining" step for the encoder.
>
> An interesting direction for incorporating realistic SCMs during training is to develop more general encoder training strategies, such as using self-supervised learning for dataset encoding. Another option is to pursue end-to-end training of Cond-FiP, or adopt curriculum learning by first training the encoder to predict  noise variables, and then fine-tuning it using the decoder's reconstruction loss on more realistic SCMs. However, we consider these extensions beyond the scope of the current work.
>
> Finally, we emphasize that while prior work on amortized causal learning has largely focused on causal effect estimation or causal discovery (as discussed in Section 5), our work tackles the novel task of amortized inference of causal mechanisms of SCMs. *We view our experiments with additive noise models as a critical proof-of-concept that demonstrates the feasibility and potential of this direction, supported by extensive evaluations and ablation studies validating our claims.*  We believe our framework can serve as a good motivation for future works that incorporate real-world datasets during training.
>
> *Inference on Real-World Datasets:* We re-emphasize that during inference, Cond-FiP does not require access to noise variables. Instead, it only needs the observations (and a predicted or true causal graph) to infer the functional mechanisms. This enables its application to real-world datasets, as demonstrated in our Sachs experiment (Appendix I).
>
> ### **Leveraging interventional data**
> ---
> We wish to clarify that the dataset encoder can incorporate interventional data during training. Given an interventional dataset and the corresponding (intervened) causal graph, our framework applies without modification. In the case of hard interventions that fix certain nodes to specific values, the prediction targets for those nodes can be set as a constant during encoder training. For interventional generation during inference, we have provided details for interventional generation in Appendix A.3.
>
> ### **Evals on more complex non-linear mechanisms**
> ---
> Thank you for this thoughtful suggestion. We agree that MLPs are a powerful and widely used tool for modeling complex nonlinear functions. In our current work, we primarily followed the AVICI setup, which uses Random Fourier Features (RFF) to introduce nonlinearity. This choice enabled a controlled and comprehensive evaluation of Cond-FIP's ability to infer nonlinear causal mechanisms from the input context (causal graph and observational data). Additionally, we evaluated Cond-FIP on the real-world Sachs dataset (Appendix I), where the true causal mechanisms are unknown and likely to be highly nonlinear. Notably, Cond-FIP achieves generalization performance on par with baselines trained from scratch on Sachs, indicating its ability to recover complex nonlinear relationships after having been trained on SCMs with RFF causal mechanisms.
>
> Following your suggestion, we further extended our evaluation to explicitly include more expressive nonlinear functions using MLPs. Specifically, we adapted the weak arrow dataset from the C-Suite benchmark, where one of the nodes has eight parents. We replaced its mechanism with a two-layer MLP with ReLU activations and weights sampled from $N(0, I)$. We ran experiments over five such random MLP parameterizations and report the mean (standard error) below.
>
> |          | Noise Pred. | Sample Gen. | Interventional Gen. |
> |----------|--------|--------|--------|
> | FiP      | 0.80 (0.06)  | 1.41 (0.1)     | 1.43 (0.08)      |
> | Cond-FiP | 0.79 (0.07)  | 1.43 (0.09)     |  1.48 (0.07)    |
>
>
> We observe that Cond-FiP remains comparable to FiP (slighly worse on interventional generation), despite FiP being trained from scratch on this MLP-based data, while Cond-FIP was trained only on linear and RFF mechanisms. While training Cond-FIP on SCMs with complex MLP based nonlinear causal mechanisms could potentially improve its performance further, our findings demonstrate that even training solely on RFF based nonlinear mechanisms enables Cond-FIP to generalize effectively to more challenging nonlinear scenarios at test time.
>
>
> ### **Training Cond-FiP with smaller context size**
> ---
> Thank you for pointing this out. We agree that, for a fair comparison in the scarce data setting, Cond-FiP should also be trained with fewer samples. To address this, we trained Cond-FiP using 50 samples (Cond-FiP (50)) and 100 samples (Cond-FiP (100)), and evaluated them on the corresponding scarce data regimes.
>
> Due to space constraints, we report results for sample generation in the challenging OOD setting (d=100 & RFF OUT).
>
> - $d_{D_{test}}= 50$
>     - Cond-FiP (50): 0.62 (0.05)
>     - Cond-FiP (400): 0.65 (0.06)
>
> - $d_{D_{test}}= 100$
>     - Cond-FiP (100): 0.65 (0.06)
>     - Cond-FiP (400): 0.63 (0.07)
>
>
> Our results show that Cond-FiP maintains performance comparable to the version trained on 400 samples, which already demonstrated strong generalization over baselines (Figure 4 & Appendix E). This suggests that even when trained with limited data, Cond-FiP continues to benefit from amortization, outperforming baselines trained from scratch under the same conditions.
>
>
> ### **Robustness to causal graph inference errors**
> ---
> We agree that access to the true causal graph at test time may not always be realistic. To address this, our experiments in the paper included a setting where the causal graph was inferred using AVICI. As shown in Figure 5 and Appendix F, Cond-FiP remained competitive with baselines trained from scratch, even without access to the ground-truth graph.
>
> To further investigate robustness, we now conduct a systematic analysis by introducing random perturbations ($p$) to the true causal graph. Specifically, we randomly remove a proportion $p$ of the true edges, such that on average $𝑝 \times$  (total edges) are missing. Due to space constraints, we report results for sample generation in the challenging OOD setting (d=100 & RFF OUT).
>
>
> |          | p=0 | p=0.01 | p=0.02 | p=0.05 | p=0.1 |
> |----------|--------|--------|--------|--------|-------|
> | FiP      | 0.55 (0.08)  | 0.55 (0.08)     |  0.57 (0.08)      |   0.62 (0.08)     |   0.68 (0.08)    |
> | Cond-FiP | 0.57 (0.07)  | 0.58 (0.07)     |  0.59 (0.07)      |   0.62 (0.07)     |   0.67 (0.07)    |
>
>
> Our findings show that Cond-FiP's performance shows significant difference after we remove 5 edges ($𝑝=0.05$). Across all tested levels of perturbation, Cond-FiP is competitive with FiP, a baseline trained from scratch for each scenario. These results demonstrate that Cond-FiP exhibits robustness to moderate errors in the input causal graph, comparable to a baseline trained from scratch. This provides more evidence to our claim that Cond-FiP learns causal mechanisms during training and can adapt to new contexts at test time by inferring functions that best explain the available information in the context (input graph and observations), even when the input causal graph is inaccurate.
>
> ### **Utility of Cond-FiP for generation tasks**
> ---
> We clarify that the baselines are trained from scratch on the the context split for each inference task, serving as the gold standard that our amortized approach aims to match. In contrast, although Cond-FiP is trained on specific scales, it generalizes well to both smaller and, more importantly, larger instance problems, while maintaining performance comparable to the baselines.
>
> The only cases where Cond-FiP performs slightly worse than the top baseline (DoWhy) are the LIN IN and LIN OUT settings for sample (Table 2) and interventional generation (Table 3). However, Cond-FiP significantly outperforms DoWhy in the RFF IN and RFF OUT settings, especially at larger scales (50 or 100 nodes). These gains are even more pronounced in the scarce data regimes (Tables 11 and 12), reinforcing Cond-FiP’s utility in non-linear settings.
>
> We believe that DoWhy can still learn reasonable solutions when the functional relationships are linear (LIN IN/OUT), making it competitive with Cond-FiP in these cases (which is still impressive as DoWhy is trained from scratch). However, learning accurate models is considerably more challenging for non-linear functional relationships (RFF IN/OUT), and Cond-FiP demonstrates a clear advantage over DoWhy.
>
> ---
> We will also correct the typos you have mentioned in the updated version of our draft. Thank you once again for your constructive comments! We are open to further discussion and would be happy to address any remaining concerns.

---

> > ### Comment · Reviewer_Zi1b · 2025-08-02
> > **Official Comment by Reviewer Zi1b**
> >
> > I appreciate the authors’ detailed response and the additional experiments. Most of my concerns have been addressed, and I maintain my current score, leaning toward acceptance.

---

> ### Author Response · Authors · 2025-08-03
>
> Thank you for taking the time to review our detailed response and additional experiments, we truly appreciate your engagement! We're glad to hear that most of your concerns have been addressed. We remain available for any further discussion or clarification you may need, and we would be grateful if you would consider increasing your score.

---

### Official Review · Reviewer_W8Yy · 2025-07-12

**Clarity:** 2
**Significance:** 3
**Originality:** 3
**Rating:** 4
**Confidence:** 2

**Summary:**

This paper employs in-context learning and a fixed-point approach to learn amortized inference of causal mechanisms for additive noise Structural Causal Models (SCMs). The authors train a single model on different SCM datasets during training time and recover the causal mechanisms from unseen input data without retraining at inference time. They perform experiments on various synthetic SCMs and the real-world Sachs dataset.

**Questions:**

Below I share my questions.

## Questions:
- How does the proposed method’s performance change as the number of training datasets increases?
- Why is positional embedding required for each dataset?
- How different can the unseen inference-time $D_X$ and $G$ be from the training data?

**Ethical Concerns:**

["NO or VERY MINOR ethics concerns only"]

**Final Justification:**

The authors addressed my concerns. Thus I raise my score to Borderline accept.

**Limitations:**

yes

**Quality:**

3

**Strengths And Weaknesses:**

Below I share my comments.

## Strengths
The paper is written in a clear and well-structured manner. The proposed model architecture and its ability to recover causal mechanisms from input data without requiring retraining, appears to be novel. I also appreciate the extensive experiments conducted on multiple synthetic datasets.


## Major Weaknesses
- (Line 104) The authors mention that they assume access to the full distribution $P_S$. How realistic is that assumption? A real-world scenario illustrating this assumption would be helpful.
- The authors mention that “Cond-FiP cannot be trained on real-world datasets since the encoder requires access to true noise variables.” This raises concerns about the practical utility of the proposed method. The authors should clarify this limitation.
- As the authors build upon the fixed-point (FiP) approach, they should provide more intuitive details about the Fip method (Section 3.2) and clarify the novel contributions of their method relative to FiP. Currently, this is unclear, especially since the FiP method also claims to design a causal generative model that learns generative fixed-point SCMs in a zero-shot manner.
- The authors state that Cond-FiP is capable of generating new data samples. Since they train on multiple SCM datasets, they should effectively learn a mixture of different SCMs. Why is this considered a novel or significant contribution? Also, what would be a practical application of this capability? This point should be addressed.

## Major Weaknesses
- It is not clearly specified how far $I'$ and $T'$ can deviate from $(T^{k})_{k=1}^{K}$.
- The introduction should be structured in a more motivational way with respect to real-world use cases. The authors mention that each new dataset requires new training, but it is unclear why this is problematic. If datasets represent significantly different distributions, retraining may be necessary anyway. To avoid retraining, how similar do the datasets need to be in terms of distribution? A concrete real-world example would help illustrate this point.
- It is unclear what the authors mean when they state that the learned SCM performs well in out-of-distribution settings. This claim needs further clarification.
- In Section 2.2 (Problem Setup), the distinction between $z$ and $D_{X}$ in Equation 3 is unclear. The role of $z$ in the learned SCM should be explained in more detail.

---

> ### Author Rebuttal · Authors · 2025-07-31
>
> We thank the reviewer for their insightful feedback. To address their comments, we first clarify the motivation and benefits of amortized inference, as this was a recurring concern, and then respond to their remaining points.
>
> ### **Why Amortized Inference?**
> ---
>
> We motivate amortized inference through the following example, consider the task of predicting the motion of objects on different planets. While trajectories vary planets across due to differences in gravitational constants, the underlying physical laws remain the same. Rather than training a new model from scratch for each planet, we can exploit this shared structure to rapidly adapt our predictions to new settings. This is the core idea behind amortized inference, leveraging common patterns across training tasks to enable fast and efficient adaptation to novel ones.
>
> Similarly, the goal of Cond-FIP is to learn a single model across multiple tasks so it can quickly adapt to new tasks at test time. Each task in our paper involves inferring the causal mechanisms of an SCM, given its corresponding causal graph and observational data.
>
> *Effectiveness of Cond-FiP:* To highlight the utility of amortized inference with Cond-FiP, note that it can infer causal mechanisms for a novel task in under one minute, while FiP takes on average 30 minutes per task as its trained from scratch. For a concrete comparison, it took us 30 hours to train Cond-FiP but we can solve each inference task in max 1 minute. Therefore, to compute our main results (Tables 1–3) we evaluated 360 different tasks, implying a total cost of $30 + 360/60 = 36$ hours with Cond-FiP.  In contrast, evaluating FIP on the same 360 tasks would require retraining from scratch each time, taking approximately $180$ hours!
>
> Thus, while Cond-FIP has a higher one-time training cost, it offers a *5× speedup over FiP* in total runtime when evaluating across multiple tasks.
>
> ### **Clarifications regarding the similarity between train and test tasks**
> ---
>
> Amortized inference relies on the presence of some shared structure across tasks. Even if a test task’s distribution differs significantly from the training tasks, this shared structure can still be leveraged for efficient generalization. In such cases, zero-shot generalization might be limited, but the amortized model may still provide a better prior that enables faster adaptation through fine-tuning, as compared to training from scratch with a randomly initialized prior.
>
> Further, we provide clarification for the following related points stated by the reviewer.
>
> > It is unclear what the authors mean when they state that the learned SCM performs well in out-of-distribution settings.
>
> In our experiments, we explicitly characterize the distribution shifts in test tasks (Appendix B.1), which arise from variations in causal graph, noise distributions, and functional mechanisms. For instance, in the OOD test tasks, we sample noise variables from a completely different distribution (Laplace) than in training tasks (Gaussian). Despite this, Cond-FIP achieves performance comparable to FiP, which is trained from scratch on each test instance. This demonstrates that Cond-FIP can zero-shot generalize to OOD instances due to its amortized training procedure.
>
> > How different can the unseen inference-time task be from the training data?
>
> To understand the limits of zero-shot generalization of Cond-FiP to OOD instances, we conducted targeted experiments (Appendix D) by systematically controlling the severity of distribution shifts, either in the causal mechanisms or in the noise distributions. We find that Cond-FIP is relatively robust to shifts in the causal mechanisms, showing minimal degradation. However, it is more sensitive to changes in the noise distribution, with performance declining as the magnitude of the shift increases.
>
> ### **Clarifications about practical utility due to access noise variables during training**
> ---
>
> *Inference on Real-World Datasets*: We agree with the reviewer that Cond-FiP cannot be trained on real-world dataset as we don't have access to their noise variables. However, it is important to note that during inference, Cond-FiP does not require access to noise samples. Instead, it only needs the observations (and a predicted or true causal graph) to infer the functional mechanisms. This allows Cond-FiP to be applied to practical real-world datasets, as demonstrated in our experiment on the Sachs dataset (Appendix I).
>
> *Future Directions for Training with Real-World Datasets:* As stated in lines 318–322, noise variables are only used as targets to train the encoder, and are not necessary for training the decoder. An interesting direction for incorporating realistic SCMs during training is to develop more general encoder training strategies, such as using self-supervised learning for dataset encoding. Another option is to pursue end-to-end training of Cond-FiP, or adopt curriculum learning by first training the encoder to predict  noise variables, and then fine-tuning it using the decoder's reconstruction loss on more realistic SCMs. However, we consider these extensions beyond the scope of the current work.
>
> Finally, we emphasize that while prior work on amortized causal learning has largely focused on causal effect estimation or causal discovery (as discussed in Section 5), our work tackles the novel task of amortized inference of causal mechanisms of SCMs. *We view our experiments with additive noise models as a critical proof-of-concept that demonstrates the feasibility and potential of this direction, supported by extensive evaluations and ablation studies validating our claims.*  We believe our framework can serve as a good motivation for future works that incorporate real-world datasets during training.
>
> ### **Clarifying the novelty of Cond-FiP over FiP**
> ---
> Thank you for your suggestion. We will update our draft to better clarify the FiP framework. Intuitively, FiP starts from an initial vector $n_0$ and simulates the fixed-point process via the DAG-attention mechanism to generate data. It is then trained using a reconstruction loss based on the observed data.
>
> However, this process is not amortized over causal mechanisms of different SCMs. The only amortization in the original FiP paper is for predicting the topological order from data, and it still needs to be trained from scratch for different tasks to infer their causal mechanisms. In contrast, our work introduces a novel modification that enables amortized learning of causal mechanisms by conditioning the fixed-point process on learned dataset embeddings. This allows Cond-FiP to generalize across tasks and infer mechanisms in a zero-shot manner. We also note that reviewers *Zi1b* and *Cobb* have independently acknowledged the novelty of our approach.
>
> Further, we provide clarification for the following related point stated by the reviewer.
>
> > Cond-FiP learns a mixture of different SCMs as its trained on multiple SCMs
>
> We clarify that Cond-FIP does not learn a mixture over the training SCMs. Instead, it conditions on a dataset-specific embedding derived from the input context (causal graph and observational data) to infer the causal mechanisms for each new task. Thus, the model adapts to each SCM individually rather than learning a mixture.
>
> ### **Other Questions**
> > How does Cond-FiP's performance change as we increase training datasets
> ---
> We analyzed the validation loss (RMSE for sample generation) across training epochs, using a total of 10k epochs, which leads to approximately 4M training datasets. The validation loss decreases sharply during the first 2k epochs and then gradually converges over the remaining 8k. This indicates that at least 0.8M training datasets (corresponding to the first 2k epochs) are needed to achieve a reasonable solution. We will include the full learning curve as a figure in the updated draft to illustrate this trend.
>
> > Authors assume access to full distribution, why is that realistic?
> ---
>
> We agree that in practice no method can be trained on instances covering the entire support of the distribution over SCMs. As noted in line 104, for the sake of simplicity in the notations we dropped the index $k$ over the finite set of SCMs $S_{k=1}^{K}$, and said assume access to the full distribution.
>
> > Distinction between $z$ and $D_X$ in E.q. 3
> ---
>
> As motivated in Section 2.1, amortized inference requires some form of context about the novel task. In our setup, $D_X$  refers to the observational data sampled from the novel SCM, which serves as part of the input context. While $z$ represents the query on which predictions are to be made. Cond-FiP uses $D_X$ to infer the causal mechanisms of the novel SCM, enabling accurate predictions for $z$.
>
> > Why are positional embeddings used?
> ---
>
> Thank you for this question! The design choice for incorporating positional embeddings in the decoder is motivated by the original FiP implementation. In FiP, the initial noise vector (which is iteratively transformed into the observation vector) is initialized as a zero vector. Consequently, its conditional embedding also remains zero, $n_0= C(D_X, G) \times 0 = 0$.
>
> To address this, we introduce positional embeddings as a bias term, i.e. $n_0= C(D_X, G) \times 0 + P(D_X, G) = P(D_X, G)$.
>
> Hence, $P(D_X, G)$ provides a learned, context-dependent initialization for $n_0$ based on the input context ($D_X, G$). Similarly, we add this positional embedding in $z_{emb}$ as well.
>
> ---
> Thank you once again for your constructive comments! We will update the introduction of our paper to better highlight the motivation behind amortized inference. We are open to further discussion and would be happy to address any remaining concerns.

---

> ### Author Response · Authors · 2025-08-04
> **Gentle Reminder**
>
> As the discussion period is nearing its end with only two days remaining, we would like to kindly draw your attention to our rebuttal, where we have addressed your concerns in detail. We would be happy to provide further clarifications should you have any additional questions.
>
> Thank you once again for your time and consideration!

---

### Note · Authors · 2025-08-12

We thank the reviewers, AC, and SAC for their time and effort in reviewing our submission. We are pleased that our rebuttal addressed the reviewers' concerns, with the final ratings of reviewers **Zi1b**, **m6sH**, and **cobb** in favor of acceptance. Since we could not get a response from the reviewer **W8Yy** before the discussion period ended, we hope our detailed rebuttal resolves their concerns and encourages them to change their stance towards acceptance as well.

We summarize our contributions for clarity.
- Prior works on amortized causal learning focus on causal discovery or effect estimation, while we tackle the novel and challenging task of amortized inference of functional mechanisms in SCMs.
- We propose a novel approach that amortizes the learning of causal mechanisms via conditional fixed point processes.
- Our method is technically sound and extensively validated for both in-distribution and OOD generalization, covering various settings with larger graphs, fewer samples, unknown causal graphs, complex noise distributions (Appendix H), and MLP nonlinear mechanisms (rebuttal response to reviewer *Zi1b*).

*Hence, we believe our work provides compelling evidence for the feasibility and utility of amortized inference of causal mechanisms in SCMs, along with a clear discussion of extensions toward training with real-world datasets.*

---
Finally, as reviewer **W8Yy** could not engage with us during the rebuttal, we summarize the key points from our response for their convenience, and we sincerely hope they consider increasing their score.

- *Why Amortized Inference?:* We provided a clear example motivating amortized inference and demonstrated its utility in terms of faster inference, as Cond-FiP provides *5 \times* speedup over its non-amortized counterpart (FiP).

- *Train–Test Task Similarity:* We clarified that complete details about the task distributions and the sampling schemes for train/test tasks are provided in our paper (App B.1). Further, our experiments also test Cond-FiP's generalization with varying levels of distribution shift (App D).

- *Novelty of Cond-FiP:* We clarified that FiP did not amortize the learning of causal mechanisms, and the proposed conditional FiP approach is the first to do so successfully. Reviewers **Zi1b** and **Cobb** independently acknowledged this novelty as well.

---

### Decision · Program_Chairs · 2025-09-17

**Decision:**

Reject

**Comment:**

The paper proposes the training of a single model on multiple datasets sampled from different additive-noise structural causal models. In-context learning and a fixed-point approach are used. Experiments on synthetic data and a real-world dataset were conducted.

While the reviewers were supportive of this paper, none of the reviewers strongly argued for acceptance. Thus I consider this a borderline paper, but I finally decided for rejection. Several aspects for improvement were mentioned during the rebuttal phase, and some of them were properly addressed by the authors. Unfortunately, some issues remain such as the assumption of access to the full distribution $\mathbb P_S$ and the requirement of access to the true noise variables, which greatly diminish the relevance and impact of the work.